# Topobexin targets the Topoisomerase II ATPase domain for beta isoform-selective inhibition and anthracycline cardioprotection

Jan Kubeš [1,8], Galina Karabanovich[2,8], Anh T. Q. Cong [3,4,8], Iuliia Melnikova[2], Olga Lenčová[5], Petra Kollárová [5], Hana Bavlovič Piskáčková [6], Veronika Keresteš[1], Lenka Applová[1], Lise C. M. Arrouye[3], Julia R. Alvey [3], Jasmina Paluncic [3], Taylor L. Witter [3,4], Anna Jirkovská[1], Jiří Kuneš[2], Petra Štěrbová-Kovaříková[6], Caroline A. Austin [7,9] ✉, Martin Štěrba[5,9] ✉, Tomáš Šimůnek[1,9] ✉, Jaroslav Roh[2,9] ✉ & Matthew J. Schellenberg [3,9] ✉

Topoisomerase II alpha and beta (TOP2A and TOP2B) isoenzymes perform essential and non-redundant cellular functions. Anthracyclines induce their potent anti-cancer effects primarily via TOP2A, but at the same time they induce a dose limiting cardiotoxicity through TOP2B. Here we describe the development of the *obex* class of TOP2 inhibitors that bind to a previously unidentified druggable pocket in the TOP2 ATPase domain to act as allosteric catalytic inhibitors by locking the ATPase domain conformation with the capability of isoform-selective inhibition. Through rational drug design we have developed topobexin, which interacts with residues that differ between TOP2A and TOP2B to provide inhibition that is both selective for TOP2B and superior to dexrazoxane. Topobexin is a potent protectant against chronic anthracycline cardiotoxicity in an animal model. This demonstration of TOP2 isoform-specific inhibition underscores the broader potential to improve drug specificity and minimize adverse effects in various medical treatments.

Topoisomerase II (TOP2) enzymes are essential regulators of DNA topology during DNA transcription, replication, and repair. In vertebrates these functions are distributed between two TOP2 isoenzymes, each with distinct cellular roles. Topoisomerase IIα (TOP2A) untangles DNA pre-catenanes during DNA replication to permit chromosome condensation and segregation in mitosis, and is primarily expressed in proliferating cells[1]. Topoisomerase IIβ (TOP2B) plays a role in transcriptional regulation and is expressed in all cell types including

[1]Department of Biochemical Sciences, Faculty of Pharmacy in Hradec Králové, Charles University; Hradec, Králové 500 03, Czech Republic. [2]Department of Organic and Bioorganic Chemistry, Faculty of Pharmacy in Hradec Králové, Charles University; Hradec, Králové 500 03, Czech Republic. [3]Department of Biochemistry and Molecular Biology, Mayo Clinic, Rochester 55905 MN, USA. [4]Mayo Clinic Graduate School of Biomedical Sciences, Mayo Clinic, Rochester 55905 MN, USA. [5]Department of Pharmacology, Faculty of Medicine in Hradec Králové, Charles University; Hradec, Králové 500 03, Czech Republic. [6]Department of Pharmaceutical Chemistry and Pharmaceutical Analysis, Faculty of Pharmacy in Hradec Králové, Charles University; Hradec, Králové 500 03, Czech Republic. [7]Biosciences Institute, Newcastle University, Newcastle upon Tyne NE1 7RU, UK. [8]These authors contributed equally: Jan Kubeš, Galina Karabanovich, Anh T. Q. Cong. [9]These authors jointly supervised this work: Caroline A. Austin, Martin Štěrba, Tomáš Šimůnek, Jaroslav Roh, Matthew J. Schellenberg. ✉e-mail: caroline.austin@newcastle.ac.uk; sterbam@lfhk.cuni.cz; simunekt@faf.cuni.cz; rohj@faf.cuni.cz; schellenberg.matthew@mayo.edu

post-mitotic cardiomyocytes[2]. Both isoforms catalyze the reaction, where one DNA duplex is passed through a transient double-strand DNA break in another DNA duplex. DNA religation can be blocked by a class of TOP2 inhibitors called "poisons", which bind to the highly conserved active site of both TOP2A and TOP2B to trap TOP2 reaction intermediates that can be processed into toxic DNA double-strand breaks[3,4].

The ability of TOP2 poisons to cause cancer cell death has led to their widespread use as chemotherapeutics. Anthracycline anticancer agents, such as daunorubicin (DAU), doxorubicin, and epirubicin are potent chemotherapeutic agents that have been used for more than 50 years to treat haematological malignancies (leukaemias, lymphomas) and solid tumours (breast cancer, sarcomas)[4,5]. However, their clinical use is hampered by their cardiotoxicity, characterized by progressive irreversible ultrastructural changes to the myocardium. These types of damage contribute to the development of cardiomyopathy and heart failure, and significantly affect cancer survivorship, especially for paediatric cancer patients[6]. While cancer cell killing is primarily mediated by the effect of anthracyclines on TOP2A[7], toxicity to non-dividing cells including cardiomyocytes is mediated by TOP2B[8]. The use of a TOP2A-specific poison or anthracycline derivative could restrict cytotoxicity to tumour cells that express high levels of TOP2A and would be highly desirable. However, any TOP2A–specific poison would need to accomplish the challenging task of discriminating between the nearly identical TOP2A and TOP2B active sites[9,10] and such a molecule with comparable potency to existing TOP2 poisons has not yet been developed.

In contrast to poisons, TOP2 catalytic inhibitors prevent ATP hydrolysis and reduce the levels of TOP2-cleaved DNA reaction intermediates, making them unavailable for interaction with anthracyclines. Co-administration of the TOP2 catalytic inhibitor dexrazoxane (ICRF-187) with anthracyclines significantly reduces the incidence of anthracycline-induced cardiotoxicity (AIC) in both animal models and in clinical settings. This cardioprotective benefit has been attributed to catalytic inhibition of TOP2B in cardiomyocytes, and the transient reduction in TOP2B catalytic activity is well tolerated[6,11]. However, dexrazoxane targets the highly conserved TOP2 dimer interface that is also identical between TOP2A and TOP2B[12–14] (Supplementary Fig. 1), fostering persisting concerns that its inhibition of TOP2A isoenzymes may interfere with anthracycline anticancer effects and augment the anthracycline-induced myelosuppression[15]. In the absence of TOP2A-specific anthracyclines, TOP2B-selective catalytic inhibitors pose a unique opportunity to strategically prevent TOP2B poisoning in the heart while leaving TOP2A in tumours susceptible to anthracycline poisoning. Here again, an isoform-selective dexrazoxane analogue would need to overcome the challenge of discriminating between identical residues in its binding sites on TOP2A and TOP2B. Therefore, a different and yet undescribed binding pocket with corresponding structurally original ligands is needed for the development of isoform-specific inhibitors with sufficient potency, selectivity, and stability necessary for use as a safe and effective cardioprotectant.

In this work we describe the development of a class of highly potent and TOP2B-selective catalytic inhibitors through rational design by building upon the general pharmacophore present in isolates from *Calophyllum Brasiliense*[16], and one of its antiproliferative derivative[17] BNS-22. Using X-ray crystallography we demonstrate that our newly developed compounds occupy a previously unidentified druggable binding-site in the TOP2 ATPase domain and function as a physical barrier to block the conformational change triggered by ATP hydrolysis. We name the class of inhibitors that target this site "*obex*", the Latin word for obstacle, to distinguish them from TOP2 inhibitors that bind at other sites. Exploiting the fact that the binding site of *obex* inhibitors contains residues that differ between the TOP2A and TOP2B isoenzymes we rationally design, prepare, and systematically evaluate a targeted panel of *obex* inhibitors aiming to develop a highly specific TOP2B inhibitor with improved drug-like properties suitable for in vivo

administration. We identify a lead molecule, namely, topobexin, as a TOP2B-specific *obex* inhibitor. We show that topobexin provides potent cardioprotective effects against AIC—both in vitro in primary cardiomyocytes and in vivo in a well-established chronic AIC rabbit model. These effects are superior to dexrazoxane not only in selectivity but also potency of TOP2B inhibition and protection of primary cardiomyocytes from anthracycline. These findings further define the TOP2B-dependence of the AIC phenomenon and support its druggability using isoform-selective *obex* inhibitors.

## Results

### TOP2 catalytic inhibition protects cardiomyocytes from daunorubicin

A derivative of the naturally occurring plant product GUT-70, BNS-22, was reported to function as a catalytic inhibitor of TOP2 similar to ICRF-193[17], which is a close bisdioxopiperazine congener of the clinically approved cardioprotectant dexrazoxane. Thus, we evaluated inhibition of TOP2 ATPase and DNA decatenation activities as well as the cardioprotective capacity of BNS-22, however, these analyses yielded disappointing outcomes. Using recombinant full-length human TOP2A and TOP2B enzymes (Supplementary Fig. 2a)[18,19] we found that BNS-22 inhibited ATP hydrolysis and DNA decatenation of TOP2 isoforms at micromolar concentrations, with an objectively negligible selectivity ratio of 1.5–1.6 in favour of TOP2B ($SR_{TOP2B}$ is defined as the TOP2A $IC_{50}$ divided by the TOP2B $IC_{50}$), similar to that of the non-selective inhibitor dexrazoxane (Supplementary Fig. 2b-e, Supplementary Table 1). Using our established primary cell model of AIC[14,20–23] that uses neonatal rat ventricular cardiomyocytes (NVCM), which express only TOP2B[24], BNS-22 demonstrated protection of NVCM against cytotoxicity induced by DAU (Supplementary Fig. 2f) at micromolar concentrations. Ultimately, we concluded that insufficient TOP2 binding, $SR_{TOP2B}$, and poor solubility in aqueous environment (limiting our evaluation $\leq 10\,\mu M$) would make BNS-22 unsuitable for cardioprotection against anthracyclines in vivo.

Therefore, we designed and evaluated a panel of molecules that build on the chromene and tetrahydroquinoline core (Fig. 1a) with three aims: To define the principal structure-activity relationships, to increase the isoform selectivity through improved binding to TOP2B, and to improve the aqueous solubility to allow in vivo assessment of druggability. We varied the constituents at position 4 and position 7 on the chromene ring (Fig. 1a) and assayed all derivatives for cardioprotection against AIC and their inherent toxicity in primary cardiomyocytes in vitro (Supplementary Figs. 3 and 4). A propyl group was the optimal constituent at position 4, providing the greatest extent of cardioprotection without the toxicity at higher concentrations that was seen with the butyl moiety. Position 7 constituents altered the relative inhibition of TOP2A vs TOP2B. Compound **2c**, with just a hydroxyl group inhibited TOP2A and TOP2B with equal efficacy (Supplementary Table 1), whereas extension of this group to a hydroxyethyl ether yielded **5c** (Fig. 1b) which brought improvement in cardioprotective effects as well as TOP2 isoform selectivity. **5c** provided significant cardioprotection from $0.1\,\mu M$ (Fig. 1c and Supplementary Fig. 3) with inherent cytotoxicity only observed at $100\,\mu M$ (Supplementary Fig. 4). **5c** is a more potent inhibitor of both TOP2A and TOP2B than BNS-22, with an improvement in selectivity toward TOP2B as determined by both the DNA decatenation assay ($SR_{TOP2B}$ 2.2, Fig. 1d) and the ATPase inhibition assay ($SR_{TOP2B}$ 3.8, Fig. 1e).

### A distinct binding pocket confers an unreported inhibition mechanism and isoform-selectivity

The potent cytoprotective effects of **5c** observed in NVCM, the trend towards TOP2B selectivity, and significantly different chemical structure of **5c** from dexrazoxane (lacking the near 2-fold symmetry of bisdioxopiperazines) all suggested that **5c** could bind to a different site in the TOP2 ATPase domain than dexrazoxane. We co-crystallized the

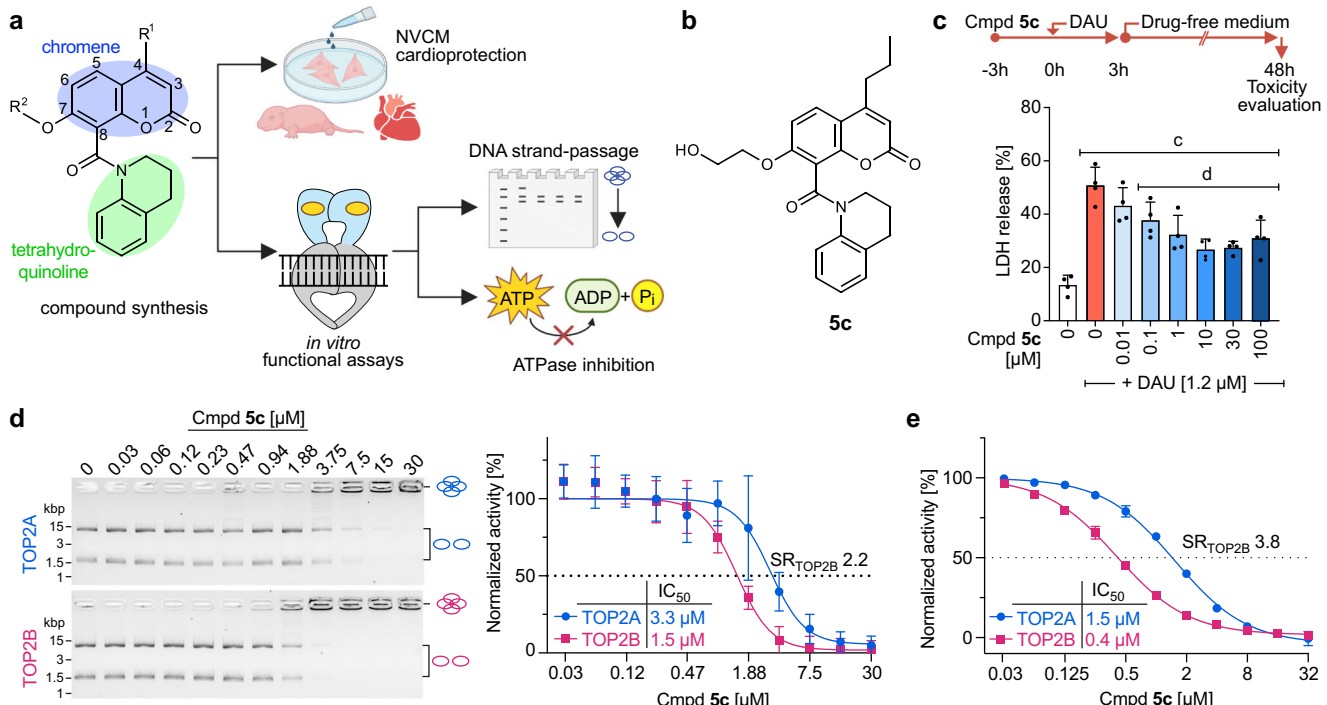

**Fig. 1 | TOP2 catalytic inhibition protects primary rat cardiomyocytes from toxicity of daunorubicin. a** General strategy for development of improved TOP2 inhibitors. Created in BioRender. Cong, A. (2025) https://BioRender.com/tncql1j. **b** Chemical structure of compound **5c**. **c** Protective effects of **5c** against toxicity (LDH release) induced by DAU [1.2 μM] in NVCM 48 h after DAU addition, $n = 4$ independent replicates, mean ± SD. Statistical significance ($P ≤ 0.05$, one-way ANOVA) against untreated cells in column 1 is indicated as (c) or DAU treated cells in column 2 indicated as (d). **d** Inhibition of decatenation activity of recombinant human TOP2A and TOP2B by **5c** ($n = 3$ independent replicates, mean ± SD normalized to untreated control). **e** Inhibition of ATPase activity of recombinant human TOP2A and TOP2B by compound **5c** ($n = 3$ independent replicates, mean ± SD normalized to untreated control). Source data are provided as a Source Data file. Precise $n$ and $P$ values for each experiment are included in Supplementary Data Table 5.

ATPase domains of TOP2A and TOP2B with **5c** and the non-hydrolysable ATP analogue AMPPNP to identify the **5c** binding site using X-ray crystallography (Supplementary Table 2). The ATPase domain of TOP2 is highly conserved between TOP2A and TOP2B (Supplementary Fig. 1) and is composed of an *N*-terminal strap, a β-hairpin that is unique to eukaryotic TOP2, a GHKL ATPase domain that binds and hydrolyses ATP, and a transducer domain that conveys mechanical motion to the DNA binding and cleavage core of TOP2[25] (Fig. 2a). Compound **5c** is clearly visible in a composite omit map in a binding site that is over 15 Å away from the nucleotide and dexrazoxane binding sites[12]. **5c** occupies the same binding pocket in both TOP2A and TOP2B, located at the nexus between the GHKL ATPase, transducer, and the 22 amino acid eukaryotic specific β-hairpin (Fig. 2b).

Compound **5c** consists of a substituted chromene core connected to a tetrahydroquinoline moiety via a non-planar amide linker (Fig. 1b). The chromene ring is buttressed between a Ser/Thr of the transducer domain and Met of the β-hairpin (Fig. 2b). The tetrahydroquinoline is accommodated within a hydrophobic pocket of the transducer domain. **5c** binding is supported by hydrogen bonding between atoms on the chromene ring and residues within this pocket (Fig. 2b and Supplementary Fig. 5a). Most of the residues interacting with the core of **5c** are similar between the two isoforms (R241, M61 and W62 for TOP2A; R257, M77 and W78 for TOP2B); however, subtle differences including a slight shift in conformation of **5c** in TOP2A relative to TOP2B are likely associated with the non-conserved S320/T336 residues and a shift in the orientation of a GHKL Tyr residue (Y82/Y98) that extrudes from the top of this binding pocket (Fig. 2c). The propyl group at position 4 is coplanar with the chromene ring in TOP2B, and orthogonal in TOP2A, but forms hydrophobic interactions with F93 and Y98 as well as the methylene of E395 (Fig. 2b and Supplementary Fig. 5b), providing a structural basis for the increased potency with

longer moieties at this position. The hydroxyethyl group at position 7 on the chromene ring extends towards W62/W78, and forms hydrogen bonds to two water molecules in both TOP2A and TOP2B (Supplementary Fig. 5c, 5d). Interestingly, **5c** induces a slight shift in Y72 of TOP2A compared to apo TOP2A due to packing of this non-conserved residue against the hydroxyethyl group, while the equivalent C88 residue in TOP2B is located 4.7 Å away and does not interact with **5c** (Supplementary Fig. 5d).

The distance of 15 Å between the compound **5c** and AMPPNP nucleotide binding sites suggests that **5c** acts via an allosteric mechanism rather than as an ATP–competitive inhibitor. We compared the TOP2B-**5c** structure to the TOP2B-ADP post-hydrolysis structure[12] (PDB ID: 7QFN) to identify the molecular basis of such inhibition. Residues surrounding the **5c** pocket are largely unchanged from those observed in its absence, suggesting the ATP-bound state can easily accommodate **5c** (Supplementary Fig. 5d). In contrast, the ADP–bound structure contains a sulphate ion in place of the phosphate released upon ATP hydrolysis and is associated with a series of movements between residues that link ATP hydrolysis to a shift of residues E395 and T336, which move into the site that is otherwise occupied by **5c** (Fig. 2d), resulting in occlusion of the **5c** pocket (Fig. 2e). When bound to this site, **5c** acts as an obstacle that would block the conformational change associated with ATP hydrolysis, which provides a mechanistic basis for TOP2 inhibition. For these reasons, we have chosen the name *obex*, which is the Latin word for obstacle, to describe the class of inhibitors that target this site. To our knowledge, no other previously described TOP2 inhibitors interact with the *obex* binding site. Moreover, it contains residues that differ between TOP2A and TOP2B (Fig. 2c) and could be leveraged to further improve the isoform selectivity of *obex* inhibitors.

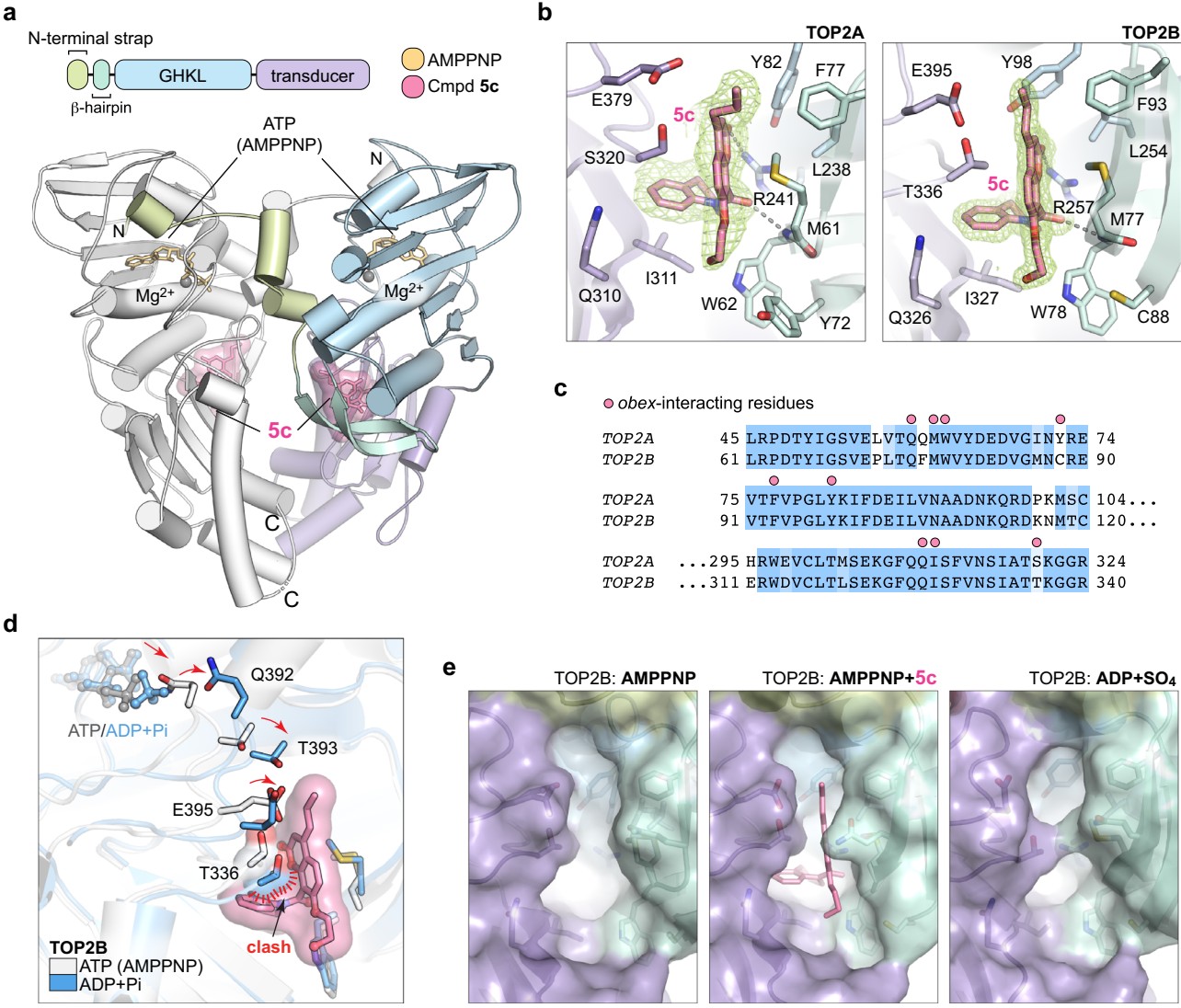

**Fig. 2 | TOP2 inhibition through binding in a newly discovered pocket that differs between isoforms. a** Overall structure of the human TOP2B ATPase bound to AMPPNP (a nonhydrolyzable ATP analogue, representing the pre-hydrolysis conformation) and compound **5c**. TOP2 ATPase subdomains are colour-coded and the second monomer coloured grey. **b** Molecular architecture of the *obex* pocket occupied by **5c** in TOP2A and TOP2B. Electron density corresponding to **5c** from a composite omit map (green mesh, contoured at 1σ) reveals the location and conformation of the compound (pink) within this pocket. **c** Sequence alignment of selected regions of the TOP2 ATPase domain. Pink circles indicate residues that interact with compound **5c**. **d** Overlay of the compound **5c** inhibited ATPase structure (containing AMPPNP) with the ATPase structure in post ATP-hydrolysis state (ADP + SO₄, PDB ID: **7QFN**). **e** Surface rendering showing the *obex* pocket of TOP2B ATPase is open in the AMPPNP (ATP)-bound form (left) but occluded in the ADP + SO₄ structure (PDB ID: **7QFN**, right). Domains are coloured as in **a**.

## Topobexin (9) is a highly selective TOP2B inhibitor

Collectively, **5c** was found to be superior to BNS-22 in all key parameters (SR$_{TOP2B}$, protection of NVCM, solubility) suggesting modification at position 7 is a viable strategy for rational development of improved *obex* inhibitors. We sought to further improve the SR$_{TOP2B}$ through interaction with Y72/C88 and reduce lipophilicity and associated toxicity at higher concentrations by generating a panel of derivatives with ionizable amine-containing moieties at position 7 (Supplementary Fig. 6). Of these, compound **9**, which contains two basic nitrogen atoms in a bulky 4-methylpiperazin-1-yl group (Fig. 3a) exhibited the most favourable properties, with no inherent toxicity observed even at 100 μM. When assayed on recombinant isoenzymes in vitro, we found that compound **9** potently inhibited TOP2B activity in the DNA decatenation assay (IC$_{50}$ = 0.19 μM), whereas its potency on TOP2A was 25 times lower (IC$_{50}$ = 4.8 μM, SR$_{TOP2B}$ 25, Fig. 3b). High selectivity toward the TOP2 isoenzyme was independently confirmed in an ATPase assay (TOP2B IC$_{50}$ = 0.35 μM, SR$_{TOP2B}$ 12; Supplementary

Fig. 7a), which is consistent with preferentially binding to the TOP2B ATPase domain. These data indicate that the methylpiperazine moiety enhances binding to TOP2B and reduces binding to TOP2A. Based on these remarkable characteristics we chose to rename compound **9** to topobexin (Topoisomerase 2B obex inhibitor).

To define the molecular basis of topobexin (**9**) isoform selectivity, we determined high-resolution crystal structures of TOP2A and TOP2B ATPase domains bound to topobexin (**9**). In TOP2B, the methylpiperazine ring expands the binding interface to include van der Waals contacts with F76, W78, and Q325. In the TOP2A structure, topobexin (**9**) forms a hydrogen bond to Y72, which is unique to the TOP2A isoenzyme, as TOP2B has C88 at the corresponding position (Fig. 3c and Supplementary Fig. 7b). Although a hydrogen bond to Y72 in TOP2A could be expected to improve binding affinity, we observe that the larger tyrosine in TOP2A shifts the methylpiperazinyl ring, which causes a rotation in the chromene core (Fig. 3d) and strains the amide bond at position 8, which has a dihedral angle of 14.3° (Fig. 3e). In an

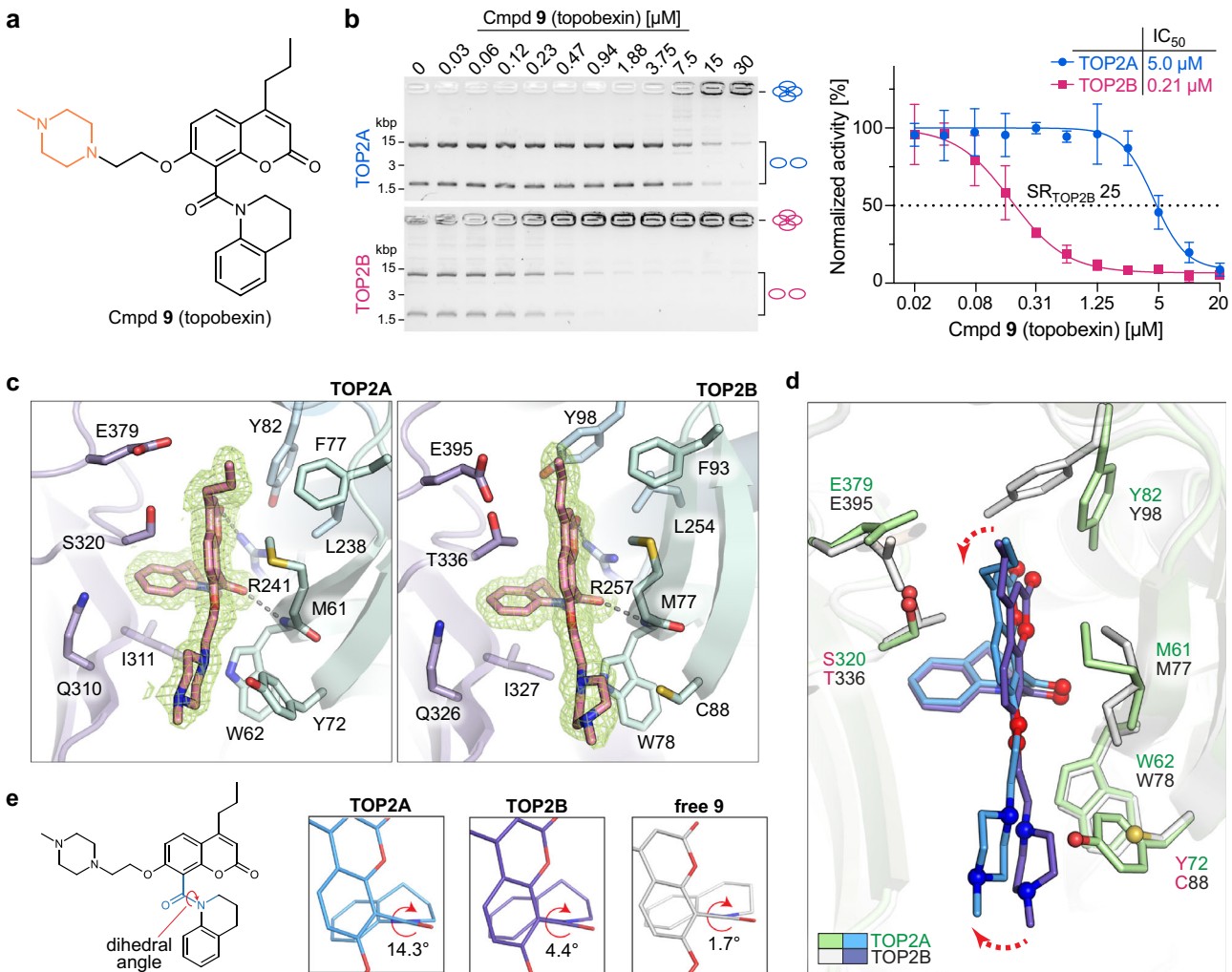

**Fig. 3 | Topobexin (9) is a highly selective TOP2B inhibitor that binds to non-identical residues in the *obex* pocket. a** Chemical structure of topobexin (**9**) (additional substitution is shown in orange). **b** Inhibition of decatenation activity of recombinant human TOP2A and TOP2B by topobexin (**9**) (*n* = 3 independent replicates, mean ± SD normalized to untreated control). **c,** Molecular architecture of the *obex* pocket binding topobexin (**9**) in TOP2A and TOP2B. Electron density corresponding to topobexin (**9**) from a composite omit map (green mesh, contoured at 1σ) reveals the location and conformation of topobexin (**9**) (pink) within this pocket. **d** Overlay of topobexin (**9**) binding in the *obex* pocket of TOP2A and TOP2B. **e** Details of the amide dihedral angle of topobexin (**9**) when bound to TOP2A, TOP2B or in unbound (free) state. Source data are provided as a Source Data file.

experimentally determined small molecule crystal structure of topobexin (**9**) (Supplementary Table 3), the amide bond has a 1.7° dihedral angle, and it is 4.4° when bound to TOP2B. In contrast, the amide dihedral angle of BNS-22 is 0.7–4° in co-crystal structures with TOP2A and TOP2B (Supplementary Fig. 7c, 7d; Supplementary Table 2), suggesting a lack of steric strain is associated with the low SR$_{TOP2B}$ of this inhibitor. Furthermore, the tetrahydroquinoline ring is shifted from that of BNS-22 in TOP2A but remains unaffected in TOP2B (Supplementary Fig. 7e, 7f). Mutation of C88 to tyrosine in TOP2B (TOP2B$^{C88Y}$) conferred significant resistance to inhibition by topobexin (**9**) but did not alter inhibition by BNS-22 (Supplementary Fig. 7g). Collectively, these data suggest steric strain induced by Y72 and the methylpiperazinyl group is the driver of isoform selectivity for topobexin (**9**).

## Selectivity of topobexin (9) and cardioprotection in cells

We next evaluated whether the marked preferential inhibition of TOP2B over TOP2A by topobexin (**9**) can also be demonstrated in a cellular context. We performed a FRAP analysis in HEK293F cells expressing fluorescently tagged TOP2 isoenzymes to evaluate whether topobexin (**9**) can selectively inhibit TOP2B in cells and immobilize TOP2B on DNA, which is indicative of closed–clamp formation, similar

to the effect of other non-poison, catalytic inhibitors of TOP2 such as ICRF-193[26]. The recovery of fluorescence for TOP2B in the photobleached area was impaired at 0.25 and 1 μM topobexin (**9**) compared to untreated control cells, while the mobility of TOP2A was not affected (Fig. 4a). In contrast, compound **2c**, which has a SR near 1 inhibits fluorescence recovery for both TOP2A and TOP2B (Supplementary Fig. 7h), suggesting that the isoform selectivity of topobexin (**9**) extends to the cellular context. Additionally, in HEK293F cells topobexin (**9**) prevented TOP2-DNA covalent complex (TOP2-DNA CC) formation induced by etoposide in an isoform-selective manner, where 1 μM of topobexin (**9**) significantly reduced TOP2B-DNA CC, while TOP2A-DNA CC was only significantly reduced at 10 μM topobexin (**9**) concentration (Fig. 4b).

Next, using NVCM we demonstrated that treatment with topobexin (**9**) provided significant protection of cardiomyocytes against DAU-induced damage at ≥ 1 μM with no significant inherent cytotoxicity of this compound alone observed up to 100 μM (Fig. 4c). The extent of protection was greater than that provided by dexrazoxane, which required at least 10× the concentration to achieve a similar extent of protection (Supplementary Fig. 8a). This agent also caused potent and concentration-dependent depletion of TOP2B protein

levels in NVCM (Fig. 4d). TOP2A was not assayed as it is not expressed in NVCM[24]. Topobexin (**9**) effectively prevented DAU-induced DNA double strand break formation in NVCM as determined by γH2AX detection even at the lowest concentration tested (0.1 μM) (Fig. 4e). The lack of γH2AX signal induced in NVCM treated with topobexin (**9**), together with the prevention of TOP2-DNA CC are consistent with the proposed mechanism of *obex* inhibitors being catalytic inhibitors (and not poisons). Furthermore, we observe that topobexin (**9**) treatment reduces DAU-induced apoptotic signaling by caspases 3/7, 8, and 9 (Fig. 4f), indicating that catalytic inhibition of TOP2B by *obex* inhibitors also mitigates the downstream signalling events and toxicity that would lead to cardiomyocyte death. Co-treatment with topobexin (**9**) also reduced the distinct changes in cardiomyocyte morphology induced by DAU (initial increased cellular volume, noticeable nuclear structure, cytoplasmic vacuolization, and granulation, eventually leading to a disrupted cellular monolayer, shrinkage of cells and nuclei, and conspicuous cell debris formation) as well as depolarization and loss of actively respiring mitochondria (Fig. 4g and Supplementary Fig. 8b). In contrast, topobexin (**9**) did not decrease the anti-proliferative effects of DAU in the TOP2A-expressing HL-60 leukaemic cell line model in any concentration tested (Supplementary Fig. 8c), and only led to a statistically significant decrease in γH2AX levels at the highest concentration tested (10 μM = approx. 2-fold TOP2A $IC_{50}$; Supplementary Fig. 8d). Collectively, these data demonstrate that topobexin (**9**) inhibits TOP2B, which reduces anthracycline-induced DNA damage, apoptosis, and cell death in post-mitotic cardiomyocytes over a range of concentrations, but does not hinder the anticancer effect of DAU in HL-60 cancer cells. Importantly, these effects correlate with the concentrations at which TOP2B is selectively inhibited over TOP2A in vitro, suggesting that beneficial, preferential inhibition extends to the cellular context.

### Topobexin (9) prevents anthracycline-induced cardiotoxicity in a rabbit model

To determine whether the cardioprotection provided by topobexin (**9**) in vitro is translatable to more complex in vivo conditions, we used a well-established non-rodent model of chronic AIC (DAU administered weekly at 3 mg/kg, i.v. to rabbits for 10 weeks), where dexrazoxane has been previously shown to be cardioprotective[11,27]. First, we established that topobexin (**9**) administration at 10 mg/kg using a 20 min i.v. infusion in rabbits provided sufficient plasma concentrations (2.5 h ≥1 μM) to allow for cardioprotective effects in vivo (Fig. 5a, $c_{max}$ = 13.5 ± 3.2 μM, $t_{max}$ = 10 min). Next, we determined that the administration of topobexin (**9**) prior to DAU can effectively prevent induction of DNA damage (evaluated as γH2AX expression) in the rabbit left ventricular (LV) myocardium 1.5 h after a single clinically relevant dose of DAU (3 mg/kg, i.e., approx. 50 mg/m²) (Fig. 5b), suggesting that topobexin (**9**) can also block anthracycline induced damage in the myocardium in vivo.

We next evaluated whether topobexin (**9**) would be cardioprotective in a chronic AIC model consisting of 10 weekly doses of DAU. Topobexin (**9**) was administered before each DAU dose in the rabbit model of chronic AIC (Fig. 5c), and compared to groups that received DAU only, topobexin (**9**) only, or saline. Topobexin (**9**) was well tolerated when administered either alone or in combination with DAU. Body weight gain (Fig. 5d) in the group receiving topobexin (**9**) alone did not differ from the control group, and the combination group matched the DAU group, suggesting no additional gross toxicity burden is caused by topobexin (**9**). While repeated administration of DAU resulted in one premature death associated with the cardiomegaly and congestion, all animals in the topobexin (**9**) + DAU group survived until the scheduled end of the experiment and necropsy revealed no apparent differences as compared to the control group. Topobexin (**9**) effectively prevented the DAU-induced decrease in LV systolic function as determined by echocardiography (Figs. 5e, f). Independent

invasive examinations of LV systolic function performed at the end of the experiment via LV catheterization confirmed very effective prevention of DAU-induced systolic dysfunction (Fig. 5g). In both echocardiographic and catheterization experiments, the indices of systolic function in the combination groups were close to saline-treated controls. LV expression of brain natriuretic peptide (BNP), which is a marker of LV wall stress and dysfunction, was significantly induced by DAU treatment yet was prevented by co-administration of topobexin (**9**) (Fig. 5h). The significant increase in cardiomyocyte damage in the DAU group as determined by plasma cTnT was again effectively prevented by topobexin (**9**) (Fig. 5i). Gene expression analysis of fibronectin 1 and collagen I alpha1 (Fig. 5j, k, respectively) revealed that DAU-induced pathological changes suggesting extracellular matrix remodeling and fibrotic changes in the myocardium that are typical of AIC were not found in the animals receiving topobexin (**9**) prior to DAU. Collectively, the proof-of-concept data presented above demonstrates that selective catalytic inhibition of TOP2B can provide effective protection from anthracycline cardiotoxicity in vivo.

## Discussion
Here we describe the discovery of a previously unidentified inhibitor binding-site located in the ATPase domain of TOP2, and a corresponding class of original TOP2 inhibitors that can occupy this site and discriminate between the alpha and beta isoforms to provide selective catalytic inhibition of TOP2B. We have named these the *obex* class of inhibitors as they act as an obstacle to the conformation change that is driven by ATP hydrolysis, which mechanistically distinguishes them from other known inhibitors of TOP2 such as the bisdioxopiperazines, ATP-competitive inhibitors, or poisons that block religation. *Obex* inhibitors function as allosteric catalytic TOP2 inhibitors at a site distinct from other catalytic TOP2 inhibitors. We have been unable to obtain crystal structures of nucleotide-free eukaryotic ATPase domains, and to our knowledge none have yet been described to date. Therefore, a comparison of the AMPPNP and AMPPNP + *obex* inhibitor structures (Supplementary Fig. 5d) reveal that the *obex* binding site is pre-formed in the ATP-bound state of TOP2, suggesting that topobexin (**9**) fits into TOP2B like a key into a lock and blocks subsequent conformational changes that could otherwise occur upon ATP hydrolysis. We have described a potent and TOP2B selective *obex* inhibitor which we have named topobexin. Furthermore, we have demonstrated that topobexin (**9**) is a cardioprotective molecule that is effective at lower concentrations than dexrazoxane in vitro, and highly effective at preventing AIC during anthracycline chemotherapy in vivo.

A potent and effective isoform-selective inhibitor has long been the "holy grail" of the topoisomerase field. All TOP2 inhibitors that have been used in the clinic or for which high-resolution structures are available target the highly conserved DNA cleavage active site (anthracyclines, etoposide, m-AMSA) or the dimer interface (dexrazoxane, ICRF-193), and these sites are comprised of residues that are invariant between the TOP2 isoforms. Here we have overcome this barrier and developed a TOP2B-selective inhibitor that targets a druggable pocket in the ATPase domain. The *obex* binding-site does not bind any previously known TOP2 ligands, and the function of this pocket appears to be to accommodate the structural changes driven by ATP hydrolysis. As such, it would not be under the same selective pressure as the active site residues, allowing amino acid residues to diverge between homologs while maintaining functionality. The structural differences in this binding pocket open the way for the development of further isoform-selective *obex* TOP2 inhibitors to preferentially inhibit other TOP2 enzymes.

TOP2B has complex functions in transcriptional regulation, including those ascribed to generating DNA breaks through scaffolding with multi-protein complexes[19,28]. The selective inhibitor topobexin (**9**) will enable experimental analysis of functions that require

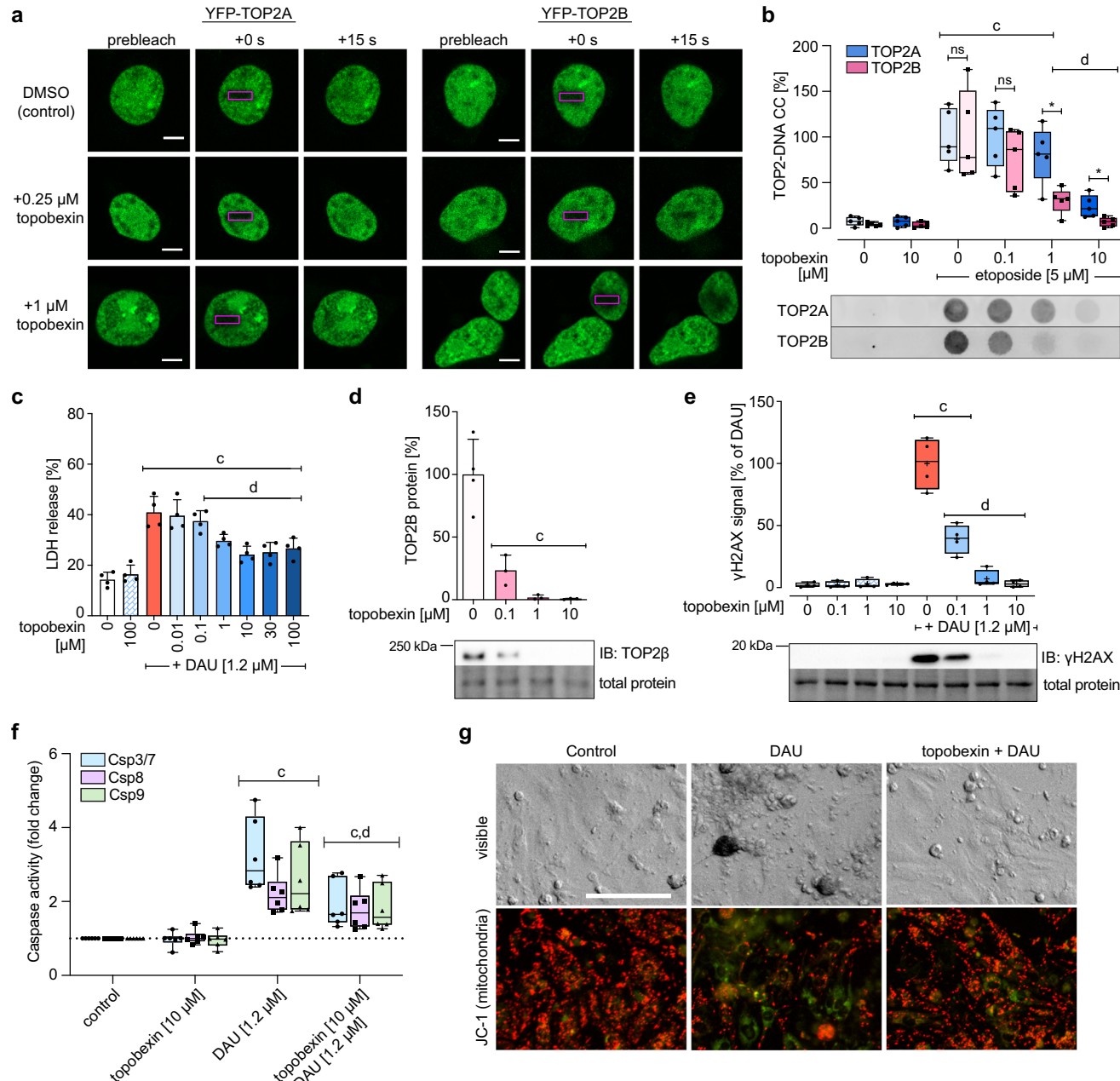

**Fig. 4 | Topobexin (9) inhibits TOP2B with high selectivity and protects cardiomyocytes in vitro. a** Fluorescence recovery after photobleaching (FRAP) analysis in HEK293F cells expressing the indicated YFP-tag protein. Bleached area is shown in a purple box, scale bar = 5 μm. **b** RADAR assays show the levels of TOP2-DNA covalent complex modulated by each TOP2 isoform in HEK293F cells evaluated by Western blotting ($n = 5$ independent replicates, $P \le 0.05$ against untreated control cells (c) or DAU (d) (one-way ANOVA)). **c** Protective effects of topobexin (9) in isolated rat neonatal ventricular cardiomyocytes (NVCM) against toxicity (LDH release) induced by DAU [1.2 μM] 48 h after DAU addition, $n = 4$ independent replicates, mean ± SD. Statistical significance ($P \le 0.05$, one-way ANOVA) against untreated cells in column 1 is indicated as (c) or DAU treated cells in column 2 indicated as (d). **d** The levels of TOP2B protein in NVCM evaluated by Western blotting ($n = 4$ independent replicates, mean ± SD, $P \le 0.05$ against untreated cells (c) (one-way ANOVA)). **e**, The levels of phosphorylated γH2AX in NVCM evaluated

by Western blotting ($n = 4$ independent replicates, $P \le 0.05$ against untreated control cells (c) or DAU (d) (one-way ANOVA)). **f** Activation of caspases after 3 h incubation with DAU and the effect of 3 h pre-treatment with topobexin (9) ($n = 6$ independent replicates, $P \le 0.05$ against untreated control cells (c) or DAU (d) (ratio paired $t$-test, two-tailed)). **g** Live-cell imaging of NVCM cells in the schedule corresponding to the cytotoxicity/protection experiments (48 h after DAU addition). Changes in morphology are shown in DIC mode while fluorescence signal from JC-1 probe corresponds to the state of mitochondria. Red signal represents actively respiring mitochondria while green indicates mitochondrial depolarization. Scale bar = 100 μm. For all box and whisker plots, center line represents the median, "+" represents the mean. Bounds of box indicate 25th to 75th percentile, whiskers indicate minimal and maximal value. Source data are provided as a Source Data file. Precise n and P values for each experiment are included in Supplementary Data Table 5.

enzyme activity separate from those that require the protein as a scaffold or component of multi protein complexes at distinct time-points in gene regulation[29,30]. Compared to methods that involve genetic manipulation, selective inhibition of TOP2B will enable rapid

inhibition at different time points, even during in vivo animal experiments. Thus, in addition to its clinical potential as a progenitor of cardioprotectants, topobexin (9) will also be an important tool for interrogating the diverse biological functions of TOP2B.

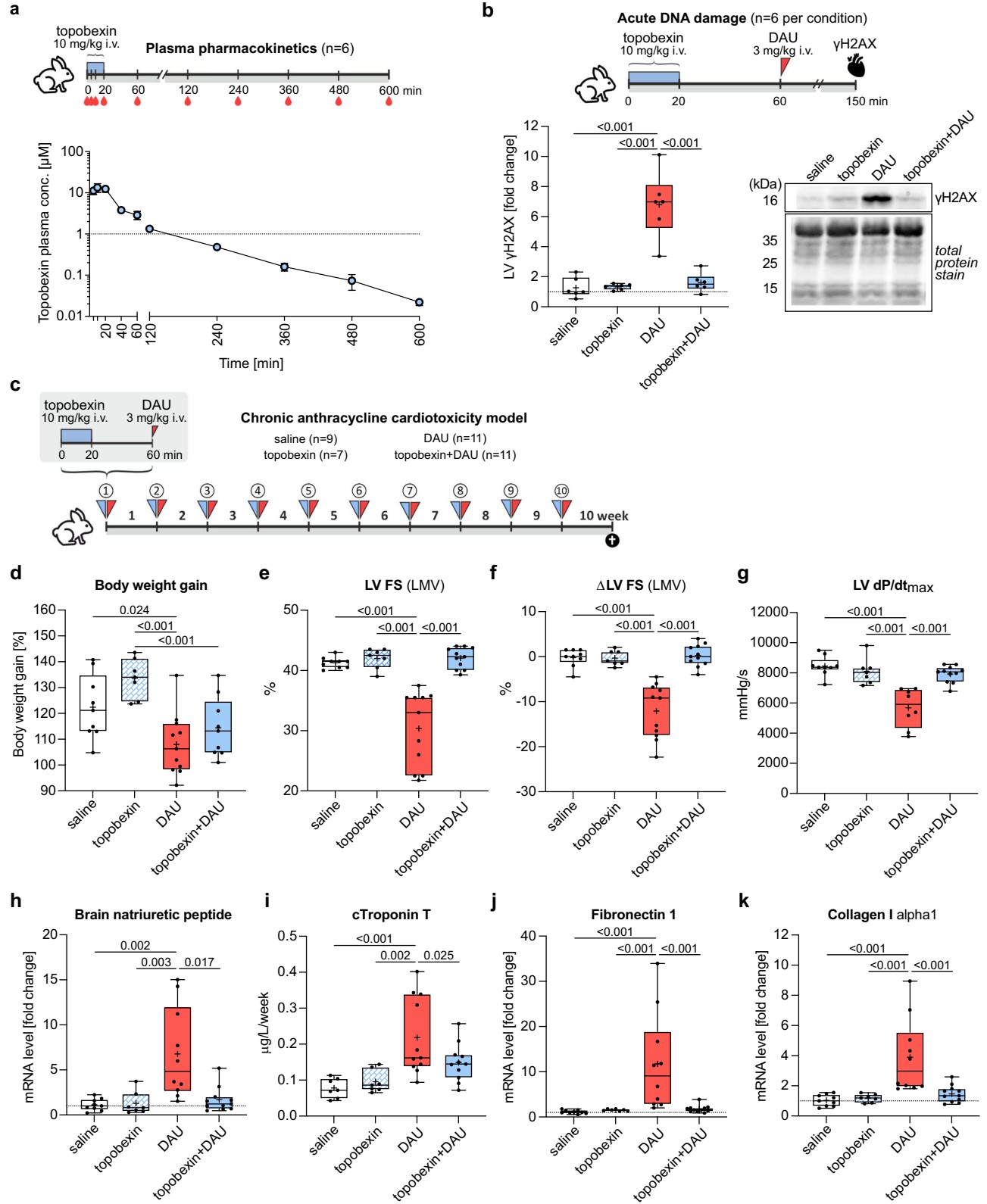

Anthracyclines are an effective and valuable component of the standard treatment regimens for cancers diagnosed in hundreds of thousands of new patients annually in the United States alone[31], but carry the risk of AIC, which may clinically manifest months or years after completion of chemotherapy. Other TOP2A/B-poisons such as the podophyllotoxin etoposide do not pose such a risk of cardiomyopathy despite sharing the same binding area in TOP2. However, in addition to TOP2B-mediated toxicities, anthracyclines may have additional toxic effects such as histone eviction[32] or greater tissue penetrance than etoposide which may contribute to the higher risk of cardiotoxicity. AIC impacts the morbidity, mortality, and quality of life in cancer survivors, who can be left in their post-cancer lives with devastating cardiomyopathy and a markedly increased risk of premature cardiovascular disease, which is of particular importance in

**Fig. 5 | Selective TOP2B inhibitor, topobexin (9), prevents anthracycline-induced cardiotoxicity in a rabbit model. a** Plasma pharmacokinetics of topobexin (**9**) after single i.v. dose administration to rabbits (10 mg/kg via 20 min infusion, *n* = 6 independent replicates) were determined using LC-MS/MS. **b**, Western blotting analysis of γH2AX in the left (LV) ventricular myocardium 1.5 h after a single administration of daunorubicin (DAU, 3 mg/kg). For the combination treatment, DAU was administered 40 min after the end of topobexin (**9**) infusion (*n* ≥ 6 independent replicates in each group). The quantitative comparisons involved multiple membranes, but all samples were processed and analyzed in parallel and the data from each membrane were normalized on the same internal standard prepared by pooling aliquots from all samples in the study. **c–k** Chronic cardiotoxicity experiment in rabbits: topobexin (**9**) and DAU were administered as in previous single dose experiments (A, B), repeated once weekly for 10 weeks and the indicated parameters were analyzed one week after the last dose. **c** Dosing regime in the chronic study. **d** Body weight gain. **e** Left ventricular fractional

shortening (LV FS) examined by echocardiography – last measured values (LMV) in all animals. **f** Treatment-induced change in LV FS (Δ LV FS) during the experiment (the difference between last measured values and values obtained at the beginning of the experiment in individual animals). **g** LV dP/dt$_{max}$ as an index of systolic function examined at the of the study via left ventricular catheterization. **h** Brain Natriuretic Peptide (BNP) mRNA levels in the LV myocardium were determined by quantitative RT-PCR. **i** Area Under Curve (AUC) of plasma concentrations of cardiac troponin T (cTnT), **j**, Fibronectin 1 and **k**, collagen I alpha-1 mRNA levels as determined by quantitative RT-PCR. *$P$ < 0.05 compared to the indicated samples or # as compared with the topobexin (**9**)-alone group, determined using one-way ANOVA. For all box and whisker plots, center line represents the median, "+" represents the mean. Bounds of box indicate 25th to 75th percentile, whiskers indicate minimal and maximal value. Source data are provided as a Source Data file. Precise n and P values for each experiment are included in Supplementary Data Table 5.

paediatric patients. Despite advances in formulation and the imposition of maximum lifetime doses, the risk of AIC remains real and significant. For many years, dexrazoxane has stood as the sole agent with a track record of effectively safeguarding against AIC, both in experimental investigations and when administered to patients. Dexrazoxane binds to a highly conserved pocket within the dimer interface of the TOP2 ATPase that is comprised of identical residues in both TOP2A and TOP2B (Supplementary Fig. 1) and inhibits ATP hydrolysis for both isoforms with equal potency (Supplementary Table 1). The possibility of TOP2A inhibition has raised persistent concerns that dexrazoxane may reduce the anticancer effectiveness of anthracyclines in cancer patients[33]. Although most studies have found no evidence for this[6,34], regrettably, whether these concerns are genuine or perceived, its utilization in clinical settings has remained limited. Topobexin (**9**) has a mechanism of action that allows for isoform-selectivity and is distinct from that of the bisdioxopiperazine drugs, which may indeed modify perception and encourage greater utilization of pharmacological cardioprotection against AIC, particularly in paediatric cancer patients that will need a lifetime of healthy heart function after cancer treatment.

Here we have demonstrated effective cardioprotection using topobexin (**9**), a highly selective and pioneering agent in the "*obex* class" of topoisomerase inhibitors, which provides the means of preferentially inhibiting the TOP2B isoform. Therefore, this discovery has established that selectively targeting TOP2B is an effective approach for cardioprotection that minimizes the risk of unintended effects on TOP2A (Fig. 6). That topobexin (**9**) can protect against AIC provides further evidence that cardioprotection is due to inhibition of TOP2B. Dexrazoxane was originally believed to protect the heart via its active metabolite ADR-925, a potent iron chelator, displacing iron from its redox active complexes with anthracyclines[35]. However, we have recently demonstrated that ADR-925 itself is ineffective, and dexrazoxane achieves cardioprotection by inhibiting and/or depleting TOP2B[11]. Our findings in this report combined with a seminal study demonstrating that specifically deleting TOP2B in cardiomyocytes protects mice from AIC development[8] underscore the central role of TOP2B in effective protection against AIC.

## Methods
### Ethics statement
All procedures in this study involving animals were in accordance with European directive 2010/63/EU (European Union, 2010) on the protection of animals used for scientific purposes and Czech Act No 246/1992 Coll. (The Czech National Council, 1992) on the protection of animals against cruelty, and were approved and supervised by the Charles University Faculty of Pharmacy Committee for Ensuring the Laboratory Animals' Welfare (Protocol MSMT - 27872/2022-3) or the Animal Welfare Committee of Charles University, Faculty of Medicine in Hradec Králové (MSMT - 1766/2021-3).

## Chemistry
Synthetic procedures and characterization of compounds are described in the Supplementary information.

## Protection against DAU-induced cardiotoxicity in NVCMs
The protective effects of the BNS-22 and newly prepared compounds were evaluated using primary cultures of neonatal rat cardiomyocytes isolated in house as previously described[14]. Hearts isolated from 1- to 3-day-old Wistar rats were minced in buffer (1.2 mM MgSO$_4$ · 7H$_2$O, 116 mM NaCl, 5.3 mM KCl, 1.13 mM NaH$_2$PO$_4$ · H$_2$O, 20 mM 4-(2-hydroxyethyl)-1-piperazine ethanesulfonic acid (HEPES), glucose 1 mg/mL) on ice, then serially digested (20 min, six repetitions) at 37 °C with collagenase type II (60 U/mL, Gibco). After 2 h pre-plating in a 150 mm Petri dish per approximately 20 hearts to minimize non-myocyte contamination, cells were plated in 24-well gelatin-coated plates at a density of 400,000 cells per well. Cells were cultured at 37 °C and 5% CO$_2$ in Dulbecco's modified Eagle medium (DMEM)/ F12 culture medium supplemented with 10% horse serum, 5% fetal bovine serum (FBS), 20 mM sodium pyruvate, 50 U/mL penicillin, and 50 µg/mL streptomycin. Freshly isolated cells were left for 40 h to attach and form a culture of spontaneously beating cardiomyocytes, then the medium was changed to DMEM/F12 medium supplemented with 5% fetal bovine serum, 20 mM sodium pyruvate, 50 U/mL penicillin, and 50 µg/mL streptomycin. For the experiments, the medium was changed to serum- and pyruvate-free DMEM/F12 with 50 U/mL penicillin and 50 µg/mL streptomycin. Cardiomyocytes were pretreated with BNS-22 or new compounds for 3 h and then coincubated with DAU (1.2 µM) for another 3 h. After the DAU treatment period, the culture medium was exchanged for the drug-free DMEM/F12 with 50 U/mL penicillin and 50 µg/mL streptomycin for the 48 h follow-up. Experiments to determine the inherent toxicity of each compound studied follow the same schedule, only no DAU was added. Subsequently, the culture medium sample was taken from each well for evaluation of lactate dehydrogenase (LDH) activity. The total LDH activity of each sample for normalization was measured after 1 h of incubation at 37 °C in lysis buffer (0.1 M potassium phosphate, 1% Triton X-100, 1 mM dithiothreitol (DTT), 2 mM EDTA, pH 7.8, 15 min in RT). All samples were immediately analyzed in Tris HCl buffer (pH 8.9) containing 35 mM lactic acid and 5 mM NAD$^+$. The rate of NAD$^+$ reduction was monitored spectrophotometrically at 340 nm for 2 min. The slope of the linear region was calculated, and the results were expressed as a percentage of the total LDH. Each presented data point was obtained from distinct sample. Statistical significance was determined using one way ANOVA followed by the Holm–Sidak post hoc test.

## Inhibition of human recombinant TOP2 isoforms expressed in human cells (HEK293F)
The human topoisomerase II decatenation assay was performed to determine the inhibitory effects of BNS-22 and new compounds as

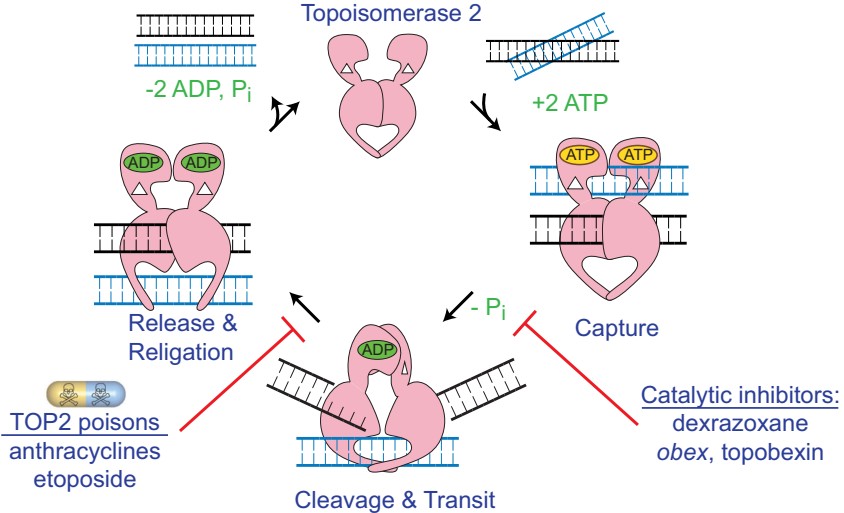

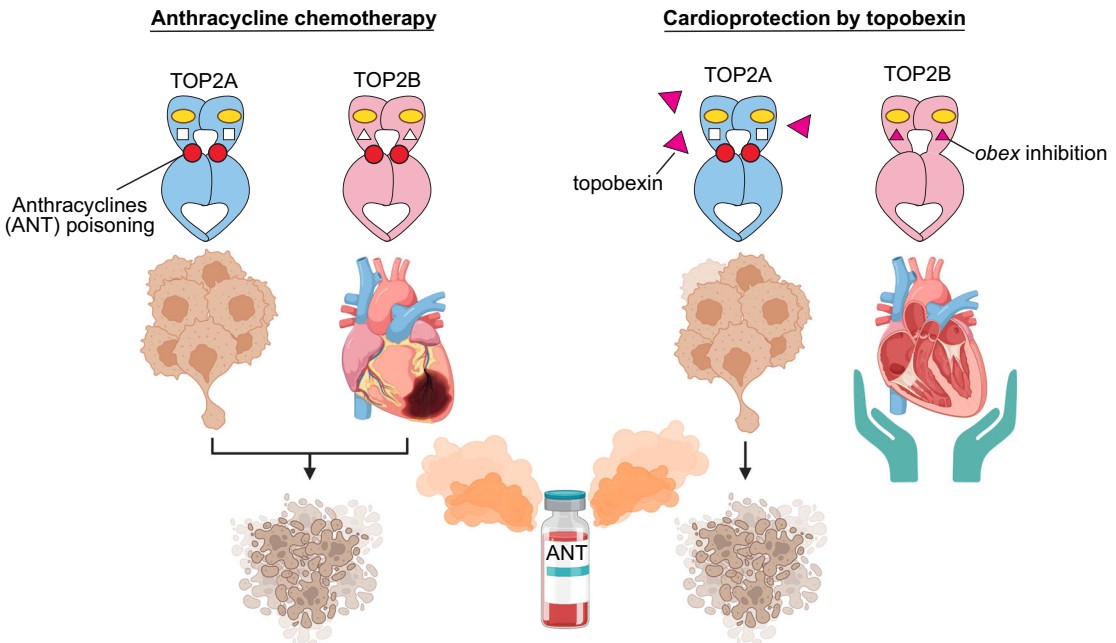

**Fig. 6 | Model of topobexin-mediated cardioprotection. a** The TOP2 reaction cycle is inhibited by the catalytic inhibitors topobexin (**9**) (TOP2B) or dexrazoxane prior to ATP hydrolysis, thus preventing the cleavage complex from forming, and trapping TOP2 in a closed-clamp conformation on DNA. TOP2 poisons anthracyclines (daunorubicin) and etoposide block the religation step. Catalytic inhibitors block the TOP2 reaction cycle prior to cleavage, and thus fewer TOP2 cleavage complexes are available for poisoning. **b** Anthracyclines poison both TOP2A and TOP2B (left) leading to cancer killing and cardiotoxicity. TOP2B can be preferentially inhibited by topobexin (**9**) which binds to the *obex* binding site on TOP2B to prevent cardiotoxicity without impairing anthracycline effects on cancer cells. Created in BioRender. Cong, A. (2025) https://BioRender.com/atckl2z.

previously described with minor modifications[14,36]. The assay was performed using recombinant human TOP2A or TOP2B generated using the HEK293F YFP-tag expression system as previously described[18,19,36]. Briefly, plasmids encoding YFP-TOP2A or YFP-TOP2B were transfected into HEK293F cells (Catalog number 11625019, Thermofisher Scientific), and TOP2 proteins were isolated from cell lysates by anti-YFP nanobody affinity chromatography. After elution by

TEV protease cleavage, TOP2 proteins were FPLC purified by sequential cation exchange and size exclusion chromatography. Plasmid encoding TOP2B C88Y was generated by round-the-horn PCR mutagenesis and the sequence was verified by full-plasmid sequencing. Purified TOP2 proteins were stored in a storage buffer containing 20 mM HEPES pH 7.5, 500 mM NaCl, 0.5 mM TCEP, 25% (v/v) glycerol at -80 °C. For decatenation assays, purified TOP2 proteins were

incubated with 200 ng kDNA (Topogen) in a reaction buffer containing 50 mM Tris−HCl (pH 8), 100 mM NaCl, 10 mM MgCl₂, 0.5 mM DTT, 2 mM ATP, 0.1 mM tris(2-carboxyethyl)phosphine (TCEP), 5% (v/v) glycerol, 70 µg/mL bovine serum albumin, and 1/10 volume of each compound diluted in 10% DMSO (final DMSO concentration 1%) for 40 min at 37 °C. The reaction was then stopped by the addition of gel loading buffer (0.5 volumes of 60% (w/v) glycerol, 60 mM EDTA, 20 mM Tris−HCl (pH 7.5), 0.5 mg/mL bromophenol blue) and the samples were placed on ice. Reaction products were resolved by agarose gel electrophoresis in TBE buffer containing GelRed stain (Biotium). Gels were visualized using an iBright CL1000 (Thermofisher Scientific). Band intensities were quantified using FIJI, and the signal of the treated samples was normalized to the value of the control (untreated sample; 100%) present on the same gel. The normalized signal of three independent measurements was then expressed as mean ± SD. Each presented data point was obtained from distinct sample. Normalized values were plotted as a function of inhibitor concentration to determine IC₅₀ values by nonlinear regression with variable slope using Prism 9 (GraphPad).

### In vitro ATPase activity of TOP2
Topoisomerase proteins were diluted to 500 nM in storage buffer containing 20 mM HEPES pH 7.5, 500 mM NaCl, 0.5 mM TCEP, 25% glycerol, and 0.2 mg/mL BSA prior to use. A coupled assay was used to measure TOP2-catalyzed ATP hydrolysis in vitro[37] in Costar 96-well plates (Corning Incorporated). 100 µL reactions contained 50 mM potassium HEPES pH 7.5, 120 mM potassium acetate, 100 mM NaCl, 8 mM magnesium acetate, 0.8 mM β-mercaptoethanol, 0.05 mg/mL BSA, 5% (v/v) glycerol, 1% (v/v) DMSO, 1.6 mM phospho(enol)pyruvate, 0.128 mM NADH, 4 U of pyruvate kinase, 6.4 U of lactate dehydrogenase, 16 ng/µL of a double-stranded DNA, 50 nM TOP2A or TOP2B, and the indicated concentration of inhibitor. Reaction plates were pre-warmed to 30 °C for 5 min and ATP was added last to initiate the reaction. The reaction plate was placed in a CLARIOstarPlus plate reader (BMG Labtech) to measure NADH fluorescence at 30 °C every 30 s over 120 cycles using a 340-60 excitation filter and a 480-100 emission filter. Background fluorescence signals calculated from wells containing no ATP were subtracted from each measurement. The slope of NADH fluorescence decay in each well was calculated using the MARS software (BMG Labtech) and reaction wells that lacked topoisomerase inhibitor to account for photobleaching of NADH fluorescence served as background subtraction. Each presented data point was obtained from distinct sample. Values were normalized to untreated controls and plotted as a function of inhibitor concentration to determine IC₅₀ values using Prism 9 (GraphPad).

### RADAR assay
HEK293F cells were plated at 3 × 10⁵ cells per well and grown overnight. The next day the wells were incubated with the indicated concentration combinations of topobexin (9) for 30 min followed by etoposide or DMSO control for 60 min (1:1000 ratio for drug dilution). Upon removal of media, cells in each well were immediately lysed with 750 µL DNAzol (Molecular Research Center, Inc.) and the DNA was precipitated by addition of 375 µL of 100% ethanol. This step was then repeated to eliminate protein aggregates. Precipitate from each sample was pelleted via centrifugation at 4 °C and max speed for 5 min. The pellets were washed twice with buffer containing 20 mM Tris pH 7.5, 1 mM EDTA pH 8.0, 50 mM NaCl, 50% EtOH, and at the end of each wash, centrifugation at 4000 × g for 2 min was repeated to isolate the DNA pellets. The pellets were washed in 75% EtOH and then collected by centrifugation at 4 °C and max speed for 5 min. The supernatant was removed via pipetting, and the remaining pellets were air dried for 5 min before 50 µL 8 mM NaOH and 0.1% Tween-20 was added to each pellet. Once no visible precipitate remained, each mixture was supplemented with 5 µL of Benzonase buffer containing 500 mM Tris pH

8.0, 10 mM MgCl₂, 1/1000 Roche complete protease inhibitor cocktail (Millipore Sigma) and 40 units Benzonase (Millipore Sigma) followed by 30 to 60 min incubation at 37 °C until the pellets were fully dissolved. The DNA content from each sample was quantified on a nanodrop, and 10 µg of DNA was diluted to a final volume of 200 µL in 8 mM NaOH and 0.1% Tween. The samples were applied over the Whatman Minifold I dot blot apparatus (Cytiva) and collected on a PVDF membrane (Thermo Fisher Scientific). The membrane was washed three times with 8 mM NaOH and 0.1% Tween-20 solution prior to removal and blocking with the same wash solution containing 1 mg/mL BSA. Western blotting was carried out on an iBind (Thermo Fisher Scientific) and TOP2 proteins were detected using isoform-specific antibody against full length TOP2A (1:250 dilution; HT2A21, Inspiralis) and TOP2B (1:500 dilution; 611493, BD Biosciences) and visualized with anti-rabbit (1:1000 dilution; 926-68071, LI-COR Bio) and anti-mouse (1:1000 dilution; 926-32210, LI-COR Bio) secondary antibodies in iBind running buffer. The blots were imaged on a LI-COR Odyssey Fc and analysed using ImageStudio. Each presented data point was obtained from distinct sample.

### Western blotting
TOP2B and γH2AX levels in NVCM were determined as follows. After incubation, 4 × 10⁵ rat neonatal cardiomyocytes were lysed with 75 µL of 2% SDS in 0.1 M Tris (pH 7.4) and boiled at 95 °C for 5 min. The protein content of each sample was determined by the BCA method and 10 µg of total protein were loaded onto a polyacrylamide gel (7.5% for TOP2B, 12% for γH2AX, Bio-Rad TGX stain-free gels). After approximately 75 min of electrophoresis (150 V, Bio-Rad Mini-PRO-TEAN Tetra Cell), the gels were transblotted onto nitrocellulose membranes using a semi-dry method in Transblot Turbo (Bio-Rad, USA). The next day, the membranes were blocked for 1 h in 5% skimmed milk powder in 0.05% TBS-T at room temperature (RT). For TOP2 analysis, membranes were incubated with rabbit anti-TOP2A/B (1:2,000, ab109524, Abcam, UK) in 1% BSA in PBS for 1 h at RT, washed 6 times for 5 min in 0.05% TBS-T and then incubated with HRP-labeled F(ab')2 goat anti-rabbit IgG (1:10,000, ab6112, Abcam, UK) in 0.05% TBS-T for 1 h RT and washed again (6 × 5 min in 0.05% TBS-T). For γH2AX analysis, membranes were incubated with mouse anti-γH2AX (1:5,000; ab26350, Abcam, UK) in 5% BSA in PBS for 1 h at RT, washed with 0.05% PBS-T for 6 × 5 min, incubated with HRP-conjugated anti-mouse IgG (1:40,000; A9044 Sigma Aldrich, USA) in 5% BSA in PBS for 1 h RT and finally washed with 0.05% PBS-T for 6 times 5 min. Target proteins were visualized using a chemiluminescent detection substrate (Clarity Western ECL Substrate, Bio-Rad, USA) and a Gel Doc EZ with Image Lab software (Bio-Rad, USA). Full, uncropped images of all blots present in the manuscript figures are included in the source data file.

For the determination of γH2aX protein in in vivo experiments, pulverized rabbit LV samples were homogenized in commercial 1× SDS Lysis Buffer (PhosphoSolutions) with phosphatase and protease inhibitor solutions (Thermo Fisher Scientific and Roche, respectively) and boiled at 95 °C for 10 min. Protein concentrations were assessed by the BCA Protein Assay Kit (Sigma-Aldrich), and 85 µg of protein were loaded into Mini-PROTEAN 12% TGX Precast Stain-free Gel (Bio-Rad). After separation, the proteins were transferred onto a nitrocellulose membrane using a Trans-Blot Turbo System (all Bio-Rad). Mouse monoclonal anti-γH2AX (1:800, ab26350, Abcam) was used as the primary antibody, and horseradish peroxidase-conjugated goat anti-mouse immunoglobulin (1:1000, P044701, DAKO) was used as the secondary antibody. The BM Chemiluminescence Blotting Substrate (Roche) and Fusion Solo S coupled with CCD camera (Vilber) were used for signal detection.

Densitometric quantification was performed using Image Lab software (Bio-Rad). Stain-free technology was used to visualize the total protein on the gel and normalize the total protein amount in each line. Each presented data point was obtained from distinct sample.

## Caspases activity determination

Caspase activity was assessed on NVCM cells using luciferase-based kits for caspase 3/7, 8 and 9 (Caspase-Glo® 3/7, 8 and 9 assay systems; Promega) according to the manufacturer's instructions on 384-well plates (Greiner Bio-One). Luminescence signal was measured using Tecan M200 multifunctional plate-reader. All measured groups were corrected for total protein concentration (BCA assay) and were expressed as fold change of the signal over corresponding untreated control group. Each presented data point was obtained from distinct sample. Statistical significance was determined using ratio paired *t*-test, two-tailed.

## Expression and purification of TOP2 ATPase

DNA encoding human TOP2A (residues 29–424) or human TOP2B (45–438) was amplified by PCR (TOP2A primers TACTTCCAATC-CAATGCATCTGTTGAAAGAATCTATCAAAAGAAAAC and TTATC-CACTTCCAAT GTTATCAGTTTAACTGGACTTGGGCC; TOP2B primers TACTTCCAATCCAATGCTTCTGTTGAGA GAGTGTATCAG and TTATC-CACTTCCAATGTTATCACTGAGTCTGAGCCTTAAATTTC) and cloned into pMCSG9[38] and transformed into BL21 Rosetta2 E. coli (EMD Biosciences). Cultures were grown in terrific broth in a LEX-48 bioreactor (Epiphyte) to an OD600 between 3 and 4. Protein expression was induced by the addition of 50 µM IPTG, 2% (v/v) ethanol, and 4% (v/v) glycerol, at 10 °C for 40 h. Cells were pelleted by centrifugation at 6000 g for 15 min and stored at -80 °C until use. Cell pellets were thawed, lysed via sonication at 4 °C in a lysis buffer (20 mM Tris pH 7.5, 300 mM NaCl, 0.5 mM TCEP, 1 mM phenylmethylsulfonyl fluoride (PMSF)) with the addition of lysozyme (0.1 mg/mL), and centrifuged at 25,000 g for 30 min to remove insoluble debris. Clarified lysate was run over Ni-NTA sepharose resin (Qiagen), washed extensively with wash buffer (20 mM Tris pH 7.5, 300 mM NaCl, 0.5 mM TCEP, 10 mM imidazole), and protein was eluted with an elution buffer (20 mM Tris pH 7.5, 300 mM NaCl, 0.5 mM TCEP, 250 mM imidazole). Protein was precipitated by mixing with 2 volumes of 4 M ammonium sulfate and immediately centrifuging at 25,000 g and 4 °C for 30 min. Protein pellet was redissolved in 1 mL of water and purified on a HiLoad 16/60 Superdex 200 column (Cytiva) with buffer containing 20 mM Tris pH 7.5, 300 mM NaCl and 0.5 mM TCEP. Fractions containing the MBP-ATPase fusion protein were identified by SDS-PAGE and pooled. TEV protease (0.02 mg/mL) was added, and the protein solution was incubated for 18 h at 4 °C to remove the His-MBP tag. TOP2 ATPase protein was further purified on a 5 mL HiTrap S-sepharose column (Cytiva) with a 0–500 mM NaCl gradient. Purified protein was concentrated to 15 mg/mL by ultrafiltration using a 10 kDa Amicon Ultra-15 Centrifugal Filter Unit (Millipore) and stored at 4 °C until use.

## Crystallization and structure determination of TOP2 ATPase

To prepare samples for crystallization, solutions of ATPase protein were supplemented with 1 mM MgCl2, 1 mM Adenylyl-imidodiphosphate (AMPPNP), and 1 mM of the indicated inhibitor, then incubated at 37 °C for 20 min. Crystals were grown using the sitting drop vapor diffusion method by mixing 200 nL concentrated protein with 200 nL precipitant using a Mosquito Xtal3 robot and IQ plates (SPT Labtech). Crystals of the TOP2α ATPase were grown in 16–23% (w/v) PEG3350, 100–300 mM NH4Cl, and 100 mM Tris pH 7.5–9. Crystals of the TOP2β ATPase were grown in 16–23% (w/v) PEG3350, 100–300 mM K3Citrate, and 100 mM HEPES pH 7–8. After 1–3 days crystals appeared in the indicated conditions and were cryoprotected prior to harvesting by washing into a cryoprotectant consisting of the crystallization condition supplemented up to 25% (w/v) PEG3350 and 15% (v/v) glycerol, and flash frozen in liquid nitrogen. X-ray diffraction data were collected at 100 K at the Advanced Photon Source (Argonne, IL, USA) 24-C and 24-E beamlines, or the Canadian Light Source (Saskatoon, SK, Canada) CMCF-ID beamline. X-ray diffraction data were processed and scaled using the HKL2000 suite[39]. Structures were solved via molecular replacement using PDB entries 1ZXM[13] or 7QFO[12] as search models using the PHENIX-PHASER[40]. Iterative rounds of manual model building using COOT[41] followed by refinement against the high-resolution datasets with PHENIX[42] produced the final models. Figures were generated using Pymol (Schrodinger Scientific).

## Small molecule crystallography

Crystals of BNS-22 were grown from a solution dissolved in 1:1 hexane:ethyl acetate by evaporation of the solvent. Crystals of the topobexin 2HCl salt were grown by dissolving 20 mg mL⁻¹ topobexin (9) in absolute ethanol and adding 1 vol of diethyl ether. Crystals appeared after 16 h at 4 °C. Crystals were dried and mounted in a cryoloop (Mitegen) in mineral oil. Crystals were harvested and flash frozen in liquid nitrogen prior to data collection at 100 K on a Synergy-R rotating Cu anode (λ = 1.5418 Å, Cu Kα) diffractometer with a Hypix Arc150 detector (Rigaku). Data were collected to a resolution of 0.8 Å, diffraction images were integrated, and scaled using CrysAlisPro (Rigaku). Structures were solved by charge flipping and refined using OLEX2[43]. Data collection and refinement statistics are described in Supplementary Table 3.

## Fluorescence recovery after photobleaching (FRAP) analysis

HEK293F cells were transfected with DNA plasmids encoding yellow-fluorescent protein (YFP)-TOP2A or YFP-TOP2B (the same DNA constructs for mammalian protein expression and purification) using Lipofectamine 2000 (Thermo Fisher Scientific). Cells were cultured for 24 h post transfection and YFP signals were confirmed prior to FRAP experiments. Transfected cells were plated on glass-bottomed culture dishes (Matsunami) and rested overnight. The next day, each dish was pre-incubated with the indicated inhibitor treatment or DMSO control for 15 min prior to FRAP experiment. Each set of experiments was repeated on at least two different days using newly transfected cells each time. Live-cell imaging was done using a LSM 980 microscope (Carl Zeiss Microscopy, LLC). YFP signal was excited at 508 nm at 2% laser power and detected through a 516–604 nm range. A small area within the nucleus (purple box) is manually selected for fluorescence bleaching at 100% laser power for 50 iterations. Inhibition of TOP2 by *obex* inhibitors was detected by impairment in fluorescence recovery at 15 s post bleaching, as a lack of inhibition allows for TOP2 diffusion throughout the nucleus.

## Live-cell microscopy and mitochondrial membrane staining

Micrographs were obtained using an Eclipse TS100 inverted epifluorescence microscope (Nikon Corporation, Tokyo, Japan) and NIS-Elements AR 2.20 software (Laboratory Imaging s.r.o., Prague, Czech Republic). After treatment and incubation time corresponding to cytotoxicity/protection experiments (3 h preincubation with topobexin (9) [10 µM], 3 h co-incubation with topobexin (9) [10 µM] + DAU [1.2 µM], 45 h drug-free medium) NVCM were stained with 0.5 µM JC-1 probe (Molecular Probes/Thermo Fisher Scientific, USA) for 30 min at 37 °C or combination of 0.2 µM MitoTracker Red CMXRos (Molecular Probes/Thermo Fisher Scientific, USA) for 30 min and Hoechst 33342 for 10 min at 37 °C and then washed with a buffer (1.2 mM MgSO₄ · 7H₂O, 116 mM NaCl, 5.3 mM KCl, 1.13 mM NaH₂PO₄ · H₂O, 20 mM HEPES).

## Animal experiments

Adult, male New Zealand White rabbits (n = 68, age: 12–16 weeks; Velaz, Czech Republic) were used for the experiments described below. The rabbits were caged individually, under standard conditions (temperature 18 °C, relative humidity: 40–50%, 12 h-long illumination period) with ad libitum access to a standard rabbit chow diet (KO-16; Velas, a.s., Czech Republic) and tap water. All animals underwent at least 2 weeks of acclimatization in the animal unit prior to randomization to study

groups. All non-invasive procedures were performed under light intramuscular anaesthesia comprising ketamine (30 mg/kg) and midazolam (1.25 mg/kg), whereas final invasive hemodynamic measurement was performed under individually titrated surgical anaesthesia (pentobarbital, approx. 10 mg/kg, i.v.) and the same was used for animal overdose.

### In vivo pharmacokinetic experiments and HPLC-MS/MS determination of topobexin (9) in plasma

Six rabbits were used to determine the plasma pharmacokinetics of topobexin (9). The compound was used as a dihydrochloride and was dissolved right before administration in sterile saline (10 mg of free base/ml); pH was optimized using NaOH and the solution was filtered through an antimicrobial filter (0.22 μm porosity, Carl Roth GmbH + Co. KG, Germany). The drug was administered i.v. using marginal ear vein at 10 mg/kg (counted as a free base) in a 20 min infusion. The dose, route of administration, and timing of infusion were determined based on preliminary pharmacokinetic experiments. Blood for pharmacokinetic analysis was collected from the central artery in the contralateral ear before drug administration (blank) and in preselected time intervals after drug administration (5 min to 10 h). Blood was collected in BD Vacutainer tubes (BD Biosciences, Plymouth, UK) containing lithium heparin. All samples were immediately centrifuged (5 min, 3000 × $g$), plasma was harvested and frozen immediately in liquid nitrogen and the samples were stored at -80 °C until analysis.

The concentration of topobexin (9) was determined in plasma samples using an analytical method developed and validated for this purpose. Briefly, rabbit plasma (50 μl) was spiked with internal standard (compound 14), precipitated with ice-cold acetonitrile (200 μl), vortexed (30 s), and centrifuged (10 min, 9520 × $g$, 4 °C). The resulting supernatant was filtered through a 0.45 μm porosity filter (Milex – Hv, Merck Milipore, Darmstadt, Germany) and analyzed using UHPLC-MS. Agilent 1290 Infinity II LC with Triple Quad LC/MS (6400 series), a Jet Stream Electrospray and Mass Hunter software (Agilent, Santa Clara, CA, USA) were used for the analyzes and data treatment, respectively. Chromatographic separation was achieved on the Zorbax Eclipse Plus C18 column (50 × 2.1 mm, 1.8 μm, Waters, Dublin, Ireland). The mobile phase composed of part A (0.0025% formic acid) and part B (acetonitrile) was mixed in the following gradient: 0–1 min (10% B), 1–3 (10–60% B), 3–3.5 min (60% B), 3.51–5 min (80% B), 5.01–6 min (10% B). The flow rate of mobile phase was 0.3 mL/min and 2 μL of the sample were injected onto the column. The linearity of the method was verified within the concentration range of 1 to 1000 nM topobexin (9) in rabbit plasma.

### Analysis of acute DAU-induced DNA damage in rabbit myocardium

Rabbits ($n = 24$) were divided equally into four groups ($n = 6$ in each) receiving a single dose as follows: DAU (3 mg/kg, i.v. bolus), saline (1 ml/kg, i.v., the control group), topobexin (9) alone (10 mg/kg, prepared and administered as describe above in the pharmacokinetic study, i.e., in 20 min i.v. infusion) and the combination group (topobexin (9) 10 mg/kg via 20 min i.v. infusion followed by DAU 3 mg/kg as i.v. bolus into the contralateral ear 40 min after the end of topobexin (9) infusion). Based on preliminary data, the experiment was completed 90 min after administration of DAU or saline. The heart was excised and γH2AX as a marker of DNA damage was determined in myocardial homogenates using Western blotting as described above.

### Chronic cardioprotective study in rabbits

A well-established model of chronic anthracycline cardiotoxicity in rabbits[27] was used to study the cardioprotective effects of topobexin (9). The model has been previously validated using clinically approved cardioprotective drug dexrazoxane, and it has been demonstrated to effectively discriminate active from non-active

derivatives of dexrazoxane and other investigational cardioprotective compounds[11,20,21,44,45].

Thirty-eight rabbits were used for this purpose using the following study design. Chronic cardiotoxicity was induced by DAU (3 mg/kg, i.v., once weekly for 10 weeks, $n = 11$), while the control group received the same volume of saline (1 ml/kg, $n = 9$) in the same way. The combination group ($n = 11$) received topobexin (9) before each of 10 DAU (weekly) administrations as described above in the acute experiments (i.e., 10 mg/kg in 20 min of i.v. infusion followed 40 min later by DAU 3 mg/kg as a bolus into the vein of the contralateral ear). An additional group of animals received topobexin (9) alone in the same manner (i.e., 10 mg/kg in 20 min i.v. infusion, once weekly for 10 weeks). Animal survival was monitored daily, while body weight change was recorded weekly. The experiment was scheduled to end a week after the last dosing of the compounds studied.

Cardiac damage induced by DAU was quantified by determination of cardiac troponin T in plasma before the 1st, 5th, 8th and 10th doses and at the end of the study using a high sensitivity assay (Elecsys Troponin T high sensitive STAT test, Roche Diagnostics, Switzerland, detection limit of 0.003 μg/L). The area under curve (AUC) of the plasma concentration of cardiac troponin T was determined using GraphPad Prism.

The systolic function of the LV was examined throughout the experiment without invasiveness using echocardiography (Vivid 4, probe 10.5 MHz, GE HeathCare) and invasively at the scheduled end of the experiment using LV catheterization (a Millar catheter 2.3 F, Millar Instruments, USA). In the echocardiography examination, the M-mode was used to determine the dimensions of the LV and the fractional shortening of the LV (LVFS) was calculated as an index of systolic function as described previously[46]. Similarly, from the change in LV pressures determined by catheterization of the LV, dP/dt$_{max}$ was calculated as an index of systolic function as previously described[47]. Statistical significance was determined using one way ANOVA followed by the Holm–Sidak post hoc test or Kruskal–Wallis ANOVA on ranks followed by Dunn's post hoc test according to the data distribution.

### Molecular markers of cardiotoxicity

During necropsy, heart was rapidly excised, washed in cold saline and LV free wall was snap frozen in liquid nitrogen. After homogenization in liquid nitrogen, aliquots (aprox. 50 mg) were stored at -80 °C and used for PCR analysis of molecular markers of cardiotoxicity. Total RNA was isolated from rabbit LV myocardium using TRIzol reagent (Sigma-Aldrich). After reverse transcription using a High-Capacity cDNA Reverse Transcription Kit, qPCR was performed with TaqMan Fast Universal PCR Master Mix on QuantStudio 7 Flex Real-Time PCR System (all from Applied Biosystems, Foster City). Primer sequences and gene accession numbers are listed in Supplementary Table 4. Mean threshold cycle values were transformed into relative expression using the Pfaffl method. Commercial gene expression assays were obtained from Applied Biosystems and Generi Biotech (Hradec Králové, Czech Republic). The expression data were normalized by *Hprt1* expression. Statistical significance was determined using one way ANOVA followed by the Holm–Sidak post hoc test or Kruskal–Wallis ANOVA on ranks followed by Dunn's post hoc test according to the data distribution.

### Antiproliferative activity and interference with the effects of DAU

Antiproliferative effects of studied compounds alone or in combination with DAU were assayed, as described previously[44], using HL-60 cell line (ATCC® Catalog Number: CCL-240TM, lot 5036502; American Type Culture Collection, Manassas, VA) derived from a patient with acute promyelocytic leukaemia. The cells were cultured in RPMI-1640 medium supplemented with 10%

FBS, 50 U/mL penicillin, and 50 µg/mL streptomycin solution at 37 °C in a humidified atmosphere of 5% $CO_2$. For the experiments, cells were seeded on 96-well plates (TPP, Switzerland) at a density of 10,000 cells per well (10,000 cells/mL). Cells were incubated with topobexin (**9**) or its combination with DAU (15 nM) for 72 h. Proliferation was assessed by the MTT assay. Briefly, 25 µL of 3 mg/mL MTT solution (3(4,5-dimethyl-2-thiazolyl)-2,5-diphenyl-2H-tetrazolium bromide) in PBS was added to each well, and after 2 h of incubation at 37 °C the cells were lysed with lysis buffer (isopropanol, 0.1 M HCl, 5% Triton X-100) for 30 minutes at RT. After the formazan was dissolved, the absorbance of the samples was measured at 570 nm and a reference wavelength at 690 nm (the 690 nm background absorbance was subtracted from the 570 nm values). Data are expressed as mean ± SD from four independent experiments. Normalized values were plotted as a function of inhibitor concentration to determine $IC_{50}$ values using Prism 9 (GraphPad). Statistical significance was evaluated by ANOVA with the Holm-Sidak post hoc test ($P \leq 0.05$, marked as significant difference against control (c) or DAU (d)).

## Statistical analysis
All statistical tests and their *P* values are summarized in Supplementary Table 5.

## Reporting summary
Further information on research design is available in the Nature Portfolio Reporting Summary linked to this article.

## Data availability
Plasmids used in this study are available upon request from the corresponding authors. Atomic coordinates and structure factors have been deposited in the PDB under accession numbers 9BQ6 (TOP2A ATPase), 9BQ7 (TOP2A ATPase + BNS-22), 9BQ9 (TOP2A ATPase + obex **5c**), 9BQB (TOP2A ATPase + topobexin), 9BQ8 (TOP2B ATPase), 9BQA (TOP2B ATPase + BNS-22), 9BQC (TOP2B ATPase + obex **5c**), and 9BQD (TOP2B ATPase + topobexin). The X-ray crystallographic coordinates for small molecule structures reported in this study have been deposited at the Cambridge Crystallographic Data Centre (CCDC), under deposition numbers 2354347 (topobexin (**9**)) and 2354203 (BNS-22). These data can be obtained free of charge from The Cambridge Crystallographic Data Centre via www.ccdc.cam.ac.uk/data_request/cif. Supplementary information is available for this paper. Source data are provided with this paper.

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

## Acknowledgements

We acknowledge Dr. Amanda Riccio (Princeton University), Dr. Ian Cowell (Newcastle University), Dr. L. Jim Maher (Mayo Clinic), and Dr. Scott Kaufmann (Mayo Clinic) for helpful comments on the manuscript. We wish to thank the NE-CAT beamline staff for their support. This research used resources of the Advanced Photon Source, a U.S. Department of Energy (DOE) Office of Science User Facility operated for the DOE Office of Science by Argonne National Laboratory under Contract No. DE-AC02-06CH11357. NE-CAT beamlines are funded by NIH-NIGMS P30 GM124165 and NIH-ORIP HEI S10OD021527 grants. Research reported in this publication was supported by the project New Technologies for Translational Research in Pharmaceutical Sciences /NETPHARM, project ID CZ.02.01.01/00/22_008/0004607 co-funded by the European Union (TS); INTER-EXCELLENCE II programme of The Ministry of Education, Youth and Sports of the Czech Republic. (Project No. LUAUS24335) (JR); Czech Science Foundation (Project No. 23-06558S) (MS); Thomas H. and Dorothy S. Corson Career Development Award in Cardiovascular Disease Research (MJS); Mayo Clinic startup funds (MJS); and Newcastle University QR ERCF People Fund (CAA).

## Author contributions

Conceptualization: M.S., T.S., J.R., M.J.S.; Methodology: J.Kub., G.K., A.T.Q.C., O.L., A.J., J.Kun., C.A.A., P.S., M.S., T.S., J.R., M.J.S.; Investigation: J.Kub., G.K., A.T.Q.C., I.M., O.L., P.K., H.B.P., V.K., L.Ap., L.Ar., J.R.A., J.P., T.L.W., A.J., J.Kun., P.S., C.A.A., M.S., M.J.S.; Funding acquisition: C.A.A., M.S., T.S., J.R., M.J.S.; Supervision: C.A.A., PS, M.S., T.S., J.R., M.J.S.; Writing – original draft: J.Kub., A.T.Q.C., C.A.A., M.S., T.S., J.R., M.J.S.; Writing – review & editing: J.Kub., G.K., A.T.Q.C., I.M., O.L., P.K., H.B.P., V.K., L.Ap., L.Ar., J.R.A., A.J., J.Kun., C.A.A., P.S., M.S., T.S., J.R., M.J.S.

## Competing interests

Obex compounds have been patented by Charles University and Mayo Clinic, with M.J.S., J.R., T.S., J.Kub, G.K., I.M., O.L., P.K., V.K., and M.S. listed as inventors (US patent application No. 63/534,074, 2023 and PCT application number PCT/US2024/043047). The remaining authors declare no competing interests.
