## [Transparent Peer Review file · Nature Communications]

Topobexin Targets the Topoisomerase II ATPase Domain for Beta Isoform-Selective Inhibition and Anthracycline Cardioprotection

Corresponding Author: Dr Matthew Schellenberg

Version 0:

Reviewer comments:

Reviewer #1

(Remarks to the Author)

The paper "Key and Lock: The First TOP2B Selective Inhibitor Topobexin Reveals a New Drug Binding Pocket for Isoform-Selective Inhibition of Topoisomerase 2 Beta and Cardioprotection Against Anthracyclines" by Jan Kubeš et al presents the development of an inhibitor, topobexin against TOP2 ATPase domain of selected isoenzyme, TOP2B. The topobexin potentially could be a potent protectant against chronic anthracycline cardiotoxicity. Overall, it is an integrated study, covering chemistry, biochemistry, structural biology, cell biology and pharmacology. It is of general interest to people in the areas from basic research to drug discovery. The report is well organized and well presented. It could be published with minor corrections. Some are listed as following:

1. "Key and Lock" in title may cause confusing. "Key and Lock" or "Lock-and-Key" is one specific protein interaction model, including some interactions between enzymes and their substrates. Generally, no significant conformational change is induced in the Lock-and-Key interaction model. In this report there is no topobexin binding structure in the absence of ANP. It is not clear if it has a Lock-and-Key interaction model or not. On the other hand, the topobexin binding to Top2, especially to Top2B, seems to be allosteric. The conformation change resulted from its binding to Top2B blocks the ATP hydrolysis in a neighboring ATP binding site. The inhibitor binding mechanism seemingly doesn't fall in a "Lock-and-Key" model. Nevertheless, it is understandable that the authors may use the term to stress the specific binding of topobexin to a selected isoenzyme, Top2B. But it is not very suitable to use the term "Key and Lock" in the description of topobexin binding.
2. All reported eight crystal structures are generally of high quality. There are still issues in some of them. In TOP2A structure (9BQ6), there is a Chain X that contains only one poorly defined water. Please remove it. In TOP2A 5c structure (9BQ9), the assignments of two magnesium ions, 907A and 908A are incorrect. There can't be any metal ions. In TOP2B 5c structure (9BQC), the model of the compound 5c is completely broken, even its electron density is clear. However, in the PDB validation report for 9BQC, 5c seems to be right. Please explain. Any manipulation?
3. In Extended Data Table 2 and Extended Data Table 3, please use the number of significant digits consistently and correctly for each entry, especially in Table 3. An error may be introduced when necessary. For example, in Table 3, the cell parameters a and b both have six significant digits while c has seven significant digits for the structure of BNS22. They are apparently overestimated. In contrast, the volume of unit cell has only five significant digits. In Table 3, "Cell Parameters" or "Cell dimensions" should be added to Column #1 to specify what those a, b, c etc are. Three cell angles should be in symbols. Also required to be explained are V and Z.
4. One missing piece of structural study is that authors did not mention any experiments on the co-crystallization of TOP2A or TOP2B with any inhibitor in the absence of ANP. These potential complex structures could provide vital information on their potential allostatic regulation of these inhibitors. Assumingly, TOP2B/Topobexin crystal is obtained first. The crystal is then used to soak ATP. A TOP2B/Topobexin/ATP structure will provide the critical evidence of the selective inhibition of the isoenzymes. A parallel experiment can be also performed with TOP2A. Author should have mentioned and discussed them.

Reviewer #2

(Remarks to the Author)

Anticancer drugs such as etoposide and anthracyclines that act by trapping the cleavage complex of human TOP2A are among World Health Organization's List of Essential Medicines. The side-effects including cardiotoxicity of these widely-available cancer drugs can be greatly reduced to provide critical improvement of the anticancer treatment if isoform selective catalytic inhibitor for human TOP2B can be identified to prevent trapping of the human TOP2B cleavage complex. The previously unidentified druggable pocket that differs between TOP2A and TOP2B for isoform specific inhibition by a new inhibitor described in this manuscript could provide highly significant advancement towards this important goal.

The manuscript describes nicely how topobexin was identified and characterized as a highly selective inhibitor that binds to the obex pocket. In general, the quality of the data encompassing medicinal chemistry and structural biology studies is high, and supports the exciting conclusion that the binding pocket occupied by topobexin can provide isoform-selective inhibition of Top2B for cardio protection against anthracyclines. The following comments should be addressed during revision of the manuscript.

1. In Figure 1c, it is not clear what the compound 5c concentrations are used for statistical comparisons c and d. Is comparison c inferring that compound 5c plus DAU still exhibit significant toxicity compared to untreated cells? Line 130 mentioned significant cardio protection from 0.01 μM . That would be comparing the second and third bars of the graph, which does not seem to correspond to either c or d. Figure 4b also needs some clarification with regard to the statistical comparisons c and d.
2. Extended Figure 8c shows that topobexin does not decrease the approximately 40% antiproliferative effect of 15 nM DAU on HL-60 cells. This is the only data presented to address the potential interference of TOP2A targeting chemotherapy by topobexin. As mentioned in Discussion, it is important not to have concern that the cardioprotective treatment may reduce the anticancer effectiveness in cancer patients. While topobexin has significantly higher IC₅₀ for inhibition of the decatenation and ATPase activity of TOP2A when compared to TOP2B and does not affect the mobility of cellular TOP2A, the manuscript has not measured the trapping of TOP2A covalent complex or induction of DNA damage in HEK293F or HL-60 cells by DAU when topobexin is present.
3. It would be helpful to have a figure of the mechanistic model of how topobexin binding to the obex pocket results in the immobilization of TOP2B on DNA as demonstrated by band depletion, as well as a conformation that reduces the formation of covalent complex during the catalytic cycle. The graphical abstract does not provide any mechanistic details of the action of topobexin.

Reviewer #3

(Remarks to the Author)

On the background of the research

The aim of the authors is to identify original drugs able to protect the cardiac muscle from the deleterious effect of anthracyclines, a major group of anticancer drugs of widespread use.

Anthracyclines have the property of stabilising the cleavable complex of topoisomerase II (TOP2) enzymes with DNA during the processes of replication and transcription.

The underlying hypothesis of the authors is that the two closely related TOP2 enzymes, TOP2A and TOP2B, which are "poisoned" by anthracyclines, play different roles and are differently located, the first one being essentially expressed in proliferative cells and involved in DNA replication, the second one being ubiquitously expressed and rather involved in DNA transcription. As a consequence, the desired pharmacological effect of anthracyclines (inhibition of cell proliferation) would be mediated by TOP2A, whereas unwanted toxic effect could be mediated by TOP2B. Indeed, the cardiac toxicity of anthracyclines, which is a major problem restricting the use of anthracyclines, especially to combat curable childhood acute leukaemia, has elicited a huge amount of research, aimed either at identifying less cardiotoxic drugs or at discovering cardiac protectors.

Many hypotheses have been formulated to explain the cardiac toxicity of anthracyclines. The most relevant one resides in the involvement of TOP2B, as emphasised by the authors; other involve the iron-mediated formation of oxygen free radicals through various mechanisms but are now discarded. We can take for granted that TOP2B poisoning by anthracyclines is the main, if not the only, mechanism of cardiac toxicity of this class of drugs.

Starting from this statement, the authors have tried to identify a drug which would take advantage of the subtle structural differences between TOP2A and TOP2B to prevent TOP2B, but not TOP2A, from interacting with anthracyclines. Such a drug, prescribed in combination with anthracyclines, would not interfere with their cytotoxic properties but would prevent their cardiotoxicity.

Two remarks: (1) The authors write "that TOP2A is essential for chromosome condensation and segregation during mitosis", whereas it is generally admitted that TOP2A is involved in DNA replication during the S phase, rather than during the M phase, in order to separate the newly synthesised DNA molecules; (2) There exist TOP2 poisons, outside the group of anthracyclines, which do not either discriminate between TOP2A and TOP2B and are devoid of cardiac toxicity; this is puzzling if TOP2B poisoning is the only mechanism involved in anthracycline cardiotoxicity. This should be briefly mentioned in the Introduction or the Discussion section.

Conception of a specific TOP2B-interacting drug

The authors have developed a major work of medicinal/pharmaceutical chemistry using 3D-structural data elaborated from the X-ray crystallographic study of the ATPase domains of TOP2A and TOP2B. Although I am not competent to judge the accuracy of their work (my skills are in pharmacology and genomics), I found the description of the results very convincing, explained with much details but very clearly, and with a number of pictures showing the interactions between candidate drugs and the catalytic sites of both TOP2A and TOP2B obtained through co-crystallography with the partner. The conclusion is that these drugs can, likely in an allosteric way, interact with TOP2B to inhibit ATP cleavage, but not with TOP2A. This rational approach brings more specificity than the empiric approach that drove to the development of dexrazoxane as a cardioprotective agent.

Pharmacological aspects of TOP2B catalytic inhibition

The differences between the modus operandi of TOP2 poisons and TOP2 catalytic inhibitors are clearly explained. The approaches used to explore the differential activities on TOP2A and TOP2B of the first compound studied ("5c", fig. 1d, e), as well as the optimised compound (topobexin, fig. 3b and ext. dat. fig. 7a) are quite relevant: ATP cleavage and DNA decatenation. (Incidentally, ext. dat. fig. 7g is labelled fig. 7f on the graph).

The use of neonatal rat ventricular cardiomyocytes (NVCM) to evaluate the cardiac toxicity of the compounds studied simply via the release of LDH is certainly a valuable approach in a first attempt and has provided interesting data. It cannot be considered as specific however, since any kind of primary cells in culture would have probably given the same results. The results obtained on NVCM cells on the protection of daunorubicin-induced H2AX formation and on caspase activity are convincing, but the comparison of dexrazoxane to topobexin (ext. dat. fig. 8a) is not convincing and the differences in LDH release are not impressive, even if significant.

For demonstrating the cardioprotective effect of a drug, an animal model, rabbit or rat, is much better than a cellular model. However, the protection against anthracycline-induced acute DNA damage (fig. 5b) is not in my view a reliable endpoint; what is important is the preservation of the contractility of the cardiac muscle on the long term, in other words the left ventricular systolic function. This has been determined by the authors by echocardiography and by systolic pressure measurements after catheterization: these two approaches are very reliable and are considered as highly predictive of cardiac toxicity (and protection) in the human clinical situation. The treatment protocol (10 weeks) appears quite satisfactory. The biochemical parameters (troponin T, BNP, etc.) are also useful but certainly less decisive than the maintenance of heart contractility.

Discussion

I have appreciated the mention that the utilization of dexrazoxane in the clinical setting has remained limited because of the persistent belief that it interfered with anthracycline efficiency. This is absolutely true and the number of children who have not benefited of dexrazoxane and have developed disabling heart failure is sadly very high. The availability of a new drug, with a mechanism of action different from that of the bisdioxopiperazine drugs, may indeed modify the perception of cardiac protection of children receiving anthracyclines.

I have no other remarks concerning this manuscript which should be published without significant modifications.

Version 1:

Reviewer comments:

Reviewer #1

(Remarks to the Author)

The Authors have adequately addressed all of my concerns and questions with the original manuscript. Therefore, I have no further comments.

Reviewer #2

(Remarks to the Author)

I am satisfied with the clarification, additional data, and new summary figure in response to my comments.

Reviewer #3

(Remarks to the Author)

The authors have kindly responded to my questions and remarks. I have no further comments about this revised version.

Nature Communications Manuscript “Key and Lock: The First TOP2B Selective Inhibitor Topobexin Reveals a New Drug Binding Pocket for Isoform-Selective Inhibition of Topoisomerase 2 Beta and Cardioprotection Against Anthracyclines”

(Manuscript # NCOMMS-24-70169)

RESPONSE TO REVIEWERS

The authors would like to thank all the three Reviewers for their thorough assessment of the manuscript and helpful comments, which were all used to make the manuscript stronger. Please see our responses and alterations below.

In addition to the changes suggested by the Reviewers, we also made modifications required by the Nature Communications author guidelines during the revision process. These include shortening the title to “Topobexin Targets a Unique Druggable Pocket of Topoisomerase II for Beta Isoform-Selective Inhibition and Anthracycline Cardioprotection”, shortening section titles, and other formatting changes to match Nature Communications article style.

Reviewer #1 (Remarks to the Author)

The paper “Key and Lock: The First TOP2B Selective Inhibitor Topobexin Reveals a New Drug Binding Pocket for Isoform-Selective Inhibition of Topoisomerase 2 Beta and Cardioprotection Against Anthracyclines” by Jan Kubeš et al presents the development of an inhibitor, topobexin against TOP2 ATPase domain of selected isoenzyme, TOP2B. The topobexin potentially could be a potent protectant against chronic anthracycline cardiotoxicity. Overall, it is an integrated study, covering chemistry, biochemistry, structural biology, cell biology and pharmacology. It is of general interest to people in the areas from basic research to drug discovery. The report is well organized and well presented. It could be published with minor corrections. Some are listed as following:

1. “Key and Lock” in title may cause confusing. “Key and Lock” or “Lock-and-Key” is one specific protein interaction model, including some interactions between enzymes and their substrates. Generally, no significant conformational change is induced in the Lock-and-Key interaction model. In this report there is no topobexin binding structure in the absence of ANP. It is not clear if it has a Lock-and-Key interaction model or not. On the other hand, the topobexin binding to Top2, especially to Top2B, seems to be allosteric. The conformation change resulted from its binding to Top2B blocks the ATP hydrolysis in a neighboring ATP binding site. The inhibitor binding mechanism seemingly doesn't fall in a “Lock-and-Key” model. Nevertheless, it is understandable that the authors may use the term to stress the specific binding of topobexin to a selected isoenzyme, Top2B. But it is not very suitable to use the term “Key and Lock” in the description of topobexin binding.

RESPONSE: We thank the Reviewer for their positive assessment of our manuscript and insightful suggestions. Indeed, an important piece of information that is missing for eukaryotic Topoisomerase 2 is a structure of the ATPase domain in the absence of bound nucleotide. Despite extensive efforts to crystallize it, neither our group nor any other has managed to do so to date. This is likely because the domain is more flexible without a bound nucleotide to constrain it to a specific conformation. However, the K_m of ATP is well below that of the cellular ATP concentration (Ling *et al.*, Structure 2022) and therefore it is likely that *obex* inhibitors would bind to a pre-engaged TOP2-ATP complex. When we compare the corresponding structures that are known: 1) ANP and 2) the corresponding ANP + inhibitor structures it becomes clear that there is no conformational change upon *obex* binding — i.e. the binding site is pre-formed. Therefore, we feel it does fall into the category of Lock-and-key interaction model (although with the caveat that this term was originally coined to describe substrate-enzyme interactions). More importantly, we chose this description to convey that in addition to the element of the pre-formed binding site, this class of inhibitors blocks a conformational change, essentially functioning by locking the conformation of Topoisomerase. Nonetheless we have removed the term “Key and Lock” from the title to avoid emphasizing this concept too much. We have also added the following statement to the discussion to convey this point:

“We have been unable to obtain crystal structures of nucleotide-free eukaryotic ATPase domains, and to our knowledge none have yet been described to date. Therefore, a comparison of the AMPPNP and AMPPNP + *obex* inhibitor structures (Supplementary Fig. 5D) reveal that the *obex* binding site is pre-formed in the ATP-bound state of TOP2, suggesting that topobexin fits into TOP2B like a key into a lock and blocks subsequent conformational changes that could otherwise occur upon ATP hydrolysis.”

2. All reported eight crystal structures are generally of high quality. There are still issues in some of them. In TOP2A structure (9BQ6), there is a Chain X that contains only one poorly defined water. Please remove it. In TOP2A 5c structure (9BQ9), the assignments of two magnesium ions, 907A and 908A are incorrect. There can't be any metal ions. In TOP2B 5c structure (9BQC), the model of the compound 5c is completely broken, even its electron density is clear. However, in the PDB validation report for 9BQC, 5c seems to be right. Please explain. Any manipulation?

RESPONSE: We thank the Reviewer for pointing these inaccuracies out. We have removed the chain X water from 9BQ6 and corrected Mg 907A and 908A to water and re-refined the structures.

We also apologize as it appears the incorrect pdb file for 9BQC was included in our submission. We have included with this resubmission the corrected coordinates (which were submitted to the PDB), where the *obex* 5c is correctly modelled.

*3. In Extended Data Table 2 and Extended Data Table 3, please use the number of significant digits consistently and correctly for each entry, especially in Table 3. An error may be introduced when necessary. For example, in Table 3, the cell parameters *a* and *b* both have six significant digits while *c* has seven significant digits for the structure of BNS22. They are apparently overestimated. In contrast, the volume of unit cell has only five significant digits. In Table 3, “Cell Parameters” or “Cell dimensions” should be added to Column #1 to specify what those *a*, *b*, *c* etc are. Three cell angles should be in symbols. Also required to be explained are *V* and *Z*.*

RESPONSE: We have changed the number of significant figures to five for the unit cell parameters and checked the statistics for consistent valid digits in tables 2 and 3. We have added the label “Cell dimensions”, and expanded “V” to “Volume” for clarification. *Z* is a standard crystallographic parameter that refers to the number of molecules per unit cell, and we have added this explanation to the revised Table 3 for crystallography non-experts.

4. One missing piece of structural study is that authors did not mention any experiments on the co-crystallization of TOP2A or TOP2B with any inhibitor in the absence of ANP. These potential complex structures could provide vital information on their potential allostatic regulation of these inhibitors. Assumably, TOP2B/Topobexin crystal is obtained first. The crystal is then used to soak ATP. A TOP2B/Topobexin/ATP structure will provide the critical evidence of the selective inhibition of the isoenzymes. A parallel experiment can be also performed with TOP2A. Author should have mentioned and discussed them.

RESPONSE: (See also our response to point 1). We concur with the Reviewer that structures of eukaryotic Topoisomerase 2 ATPase domain in the absence of bound nucleotide would be of value. However, despite extensive efforts to crystallize it, neither our group nor any other has managed to do so to date. We speculate this is because the ATPase domain is too dynamic to crystallize in the absence of nucleotide. Our attempts to co-crystallize the ATPase domain with ATP yield structures with ADP due to hydrolysis by the enzyme. Therefore, we have generated co-crystals with ANP, which occupies the same conformational state as ATP. We now mention this in the discussion (edit copied from response to point 1):

“We have been unable to obtain crystal structures of nucleotide-free eukaryotic ATPase domains, and to our knowledge none have yet been described to date. Therefore, a comparison of the AMPPNP and AMPPNP + *obex* inhibitor structures (Supplementary Fig. 5D) reveal that the *obex* binding site is pre-formed in the ATP-bound state of TOP2, suggesting that topobexin fits into TOP2B like a key into a lock and blocks subsequent conformational changes that could otherwise occur upon ATP hydrolysis.”

Reviewer #2 (Remarks to the Author)

Anticancer drugs such as etoposide and anthracyclines that act by trapping the cleavage complex of human TOP2A are among World Health Organization's List of Essential Medicines. The side-effects including cardiotoxicity of these widely-available cancer drugs can be greatly reduced to provide critical improvement of the anticancer treatment if isoform selective catalytic inhibitor for human TOP2B can be identified to prevent trapping of the human TOP2B cleavage complex. The previously unidentified druggable pocket that differs between TOP2A and TOP2B for isoform specific inhibition by a new inhibitor described in this manuscript could provide highly significant advancement towards this important goal.

*The manuscript describes nicely how topobexin was identified and characterized as a highly selective inhibitor that binds to the *obex* pocket. In general, the quality of the data encompassing medicinal chemistry and structural biology studies is high, and supports the exciting conclusion that the binding pocket occupied by topobexin can provide isoform-selective inhibition of Top2B for cardio protection against anthracyclines. The following comments should be addressed during revision of the manuscript.*

1. In Figure 1c, it is not clear what the compound 5c concentrations are used for statistical comparisons c and d. Is comparison c inferring that compound 5c plus DAU still exhibit significant toxicity compared to untreated cells? Line 130 mentioned significant cardio protection from 0.01 μ M. That would be comparing the second and third bars of the graph, which does not seem to correspond to either c or d. Figure 4b also needs some clarification with regard to the statistical comparisons c and d.

RESPONSE: We thank the Reviewer for bringing these ambiguities to our attention. We use the symbols (c) and (d) to indicate statistical significance against control and DAU treated cells, respectively, to facilitate interpretation. Denoting this information using the classical lines and “*”s makes a complex diagram that

is difficult to interpret. In the Figure 1c (and other panels with similar combination treatments), the symbol “c” above columns 2–8 shows statistically significant differences against untreated control cells (the first column, in white), which indicates that the observed toxicity for cells treated with DAU only or combination treatment of DAU with compound **5c** is higher than in control untreated cells. The symbol “d” above columns 4–8 (compound **5c** 0.1–100 μM in combination with DAU 1.2 μM) indicates statistically significant difference against DAU-treated cells (second column, red) and indicates the DAU toxicity is significantly reduced. We have revised the figure 1c legend to add column references and aid interpretability:

“Statistical significance ($P \leq 0.05$, one-way ANOVA) against untreated control cells in column 1 is indicated as (c) or DAU-treated cells in column 2 indicated as (d)”

We also list all individual P values for each comparison and statistical tests from this study in Supplementary Table 5.

Also, please note that Line 130 mentions “**5c** provided significant cardioprotection from **0.1 μM ”, which in this case would mean comparing the second and fourth bars.**

2. Extended Figure 8c shows that topobexin does not decrease the approximately 40% antiproliferative effect of 15 nM DAU on HL-60 cells. This is the only data presented to address the potential interference of TOP2A targeting chemotherapy by topobexin. As mentioned in Discussion, it is important not to have concern that the cardioprotective treatment may reduce the anticancer effectiveness in cancer patients. While topobexin has significantly higher IC50 for inhibition of the decatenation and ATPase activity of TOP2A when compared to TOP2B and does not affect the mobility of cellular TOP2A, the manuscript has not measured the trapping of TOP2A covalent complex or induction of DNA damage in HEK293F or HL-60 cells by DAU when topobexin is present.

RESPONSE: To provide additional support we have added data describing the differential inhibition of TOP2A/B covalent trapping by topobexin using the RADAR assay. Anthracyclines yield a very low signal in this assay, so we used etoposide, which is another drug that is a potent inducer of TOP2 covalent complexes and binds to the same area as anthracyclines. We assayed the effect of topobexin on etoposide-induced TOP2 covalent complexes in HEK293F cells and include this data as Fig. 4b. Here we observe a reduction of TOP2B covalent complexes at lower concentrations of topobexin than is required for TOP2A, which is in line with the in vitro enzyme assays (Fig. 3b,c; Supplementary Fig. 7a) and the concentrations present in rabbits during daunorubicin administration ($\leq 3 \mu\text{M}$, Fig. 5a). We have added the following description to the text:

“Additionally, in HEK293F cells topobexin (**9**) prevented TOP2-DNA covalent complex (TOP2-DNA CC) formation induced by etoposide in an isoform-selective manner, where 1 μM of topobexin (**9**) significantly reduced TOP2B-DNA CC, while TOP2A-DNA CC was only significantly reduced at 10 μM topobexin (**9**) concentration (Fig. 4b).”

We have data that evaluates topobexin-mediated protection against induction of DNA damage caused by DAU as measured by γH2AX in HL60 as Supplementary Fig. 8d. This experiment shows that a statistically significant reduction in γH2AX formation in these two proliferating cell lines occurs at concentrations of topobexin where TOP2A inhibition becomes significant (Supplementary Fig. 8d), in sharp contrast to

NVCMs where a pronounced reduction in γ H2AX occurs even at 0.1 μ M (Fig. 4e). Again, significant protective effects are not observed at lower concentrations where TOP2A is not inhibited, which is in line with the enzyme assay data (Fig. 3b,c; Supplementary Fig. 7a) and the concentrations present in rabbits during daunorubicin administration ($\leq 3 \mu$ M, Fig. 5a). We have added the following text to the manuscript:

“In contrast, topobexin (9) did not decrease the antiproliferative effects of DAU in the TOP2A-expressing HL60 leukemic cell line model in any concentration tested (Supplementary Fig. 8c), and only led to a statistically significant decrease in γ H2AX levels at the highest concentration tested (10 μ M = approx. 2-fold TOP2A IC₅₀; Supplementary Fig. 8d). Collectively, these data demonstrate that topobexin (9) inhibits TOP2B, which reduces anthracycline-induced DNA damage, apoptosis, and cell death in post-mitotic cardiomyocytes over a range of concentrations, but does not hinder the anticancer effect of DAU in HL-60 cancer cells. Importantly, these effects correlate with the concentrations at which TOP2B is selectively inhibited over TOP2A *in vitro*, suggesting that beneficial, preferential inhibition extends to the cellular context.”

3. It would be helpful to have a figure of the mechanistic model of how topobexin binding to the obex pocket results in the immobilization of TOP2B on DNA as demonstrated by band depletion, as well as a conformation that reduces the formation of covalent complex during the catalytic cycle. The graphical abstract does not provide any mechanistic details of the action of topobexin.

RESPONSE: We have moved the graphical abstract to figure panel 6b, where it serves as a summary of the mechanistic basis for cardioprotection by topobexin. We have also added a panel that describes the reaction cycle of TOP2 and inhibition by topobexin and anthracyclines that can be used to define the mechanistic model of TOP2 immobilization on DNA by topobexin.

Reviewer #3 (Remarks to the Author):

On the background of the research

The aim of the authors is to identify original drugs able to protect the cardiac muscle from the deleterious effect of anthracyclines, a major group of anticancer drugs of widespread use. Anthracyclines have the property of stabilising the cleavable complex of topoisomerase II (TOP2) enzymes with DNA during the processes of replication and transcription. The underlying hypothesis of the authors is that the two closely related TOP2 enzymes, TOP2A and TOP2B, which are “poisoned” by anthracyclines, play different roles and are differently located, the first one being essentially expressed in proliferative cells and involved in DNA replication, the second one being ubiquitously expressed and rather involved in DNA transcription. As a consequence, the desired pharmacological effect of anthracyclines (inhibition of cell proliferation) would be mediated by TOP2A, whereas unwanted toxic effect could be mediated by TOP2B. Indeed, the cardiac toxicity of anthracyclines, which is a major problem restricting the use of anthracyclines, especially to combat curable childhood acute leukaemia, has elicited a huge amount of research, aimed either at identifying less cardiotoxic drugs or at discovering cardiac protectors.

Many hypotheses have been formulated to explain the cardiac toxicity of anthracyclines. The most relevant one resides in the involvement of TOP2B, as emphasised by the authors; other involve the iron-mediated formation of oxygen free radicals through various mechanisms but are now discarded. We can take for granted that TOP2B poisoning by anthracyclines is the main, if not the only, mechanism of cardiac toxicity of this class of drugs.

Starting from this statement, the authors have tried to identify a drug which would take advantage of the subtle structural differences between TOP2A and TOP2B to prevent TOP2B, but not TOP2A, from interacting with anthracyclines. Such a drug, prescribed in combination with anthracyclines, would not interfere with their cytotoxic properties but would prevent their cardiotoxicity.

Two remarks: (1) The authors write “that TOP2A is essential for chromosome condensation and segregation during mitosis”, whereas it is generally admitted that TOP2A is involved in DNA replication during the S phase, rather than during the M phase, in order to separate the newly synthesised DNA molecules; (2) There exist TOP2 poisons, outside the group of anthracyclines, which do not either discriminate between TOP2A and TOP2B and are devoid of cardiac toxicity; this is puzzling if TOP2B poisoning is the only mechanism involved in anthracycline cardiotoxicity. This should be briefly mentioned in the Introduction or the Discussion section.

RESPONSE:

1) We agree that this statement could be more precise and specific and have revised this sentence to: “Topoisomerase II α (TOP2A) is essential for separating DNA pre-catenanes during DNA replication to permit chromosome condensation and segregation during mitosis and primarily expressed in proliferating cells”

2) This is a good point and we want to thank the Reviewer for raising this issue. It is true, that in contrast to anthracyclines, some other chemotherapeutics that target TOP2A/B, such as etoposide, do not pose such a risk of cardiotoxicity (they commonly do not induce cumulative dose-dependent cardiomyopathy with typical histopathological hallmarks and chronic heart failure). However, although they act on TOP2B via a similar mechanism, there are important differences between anthracyclines and etoposide. Particularly, unlike etoposide, anthracyclines have several important additional mechanisms which may be essential co-requisites for cardiotoxicity induction. For example, the group of prof. Neefjes (Qiao *et al.* PNAS 2020) has shown that such co-requisite could be anthracycline-induced chromatin damage resulting in histone eviction. This view could explain why etoposide does not share the same risk of cardiotoxicity with anthracyclines, and at the same time it is compatible with the TOP2B dependent mechanism of cardioprotection of topobexin. Indeed, other yet unexplored pharmacodynamic or pharmacokinetic dissimilarities, such as the extent of accumulation of anthracyclines in cardiac tissue or interactions with cardiolipin (Goormaghtigh *et al.* Bioc. Pharm. 1980) could be also involved. We have added the following text to the discussion to convey this point:

“Other TOP2A/B-poisons such as the podophyllotoxin etoposide do not pose such a risk of cardiomyopathy despite sharing the same binding area in TOP2. However, in addition to TOP2B-mediated toxicities, anthracyclines may have additional toxic effects such as histone eviction³² or greater tissue penetrance than etoposide which may contribute to the higher risk of cardiotoxicity.”

Conception of a specific TOP2B-interacting drug

The authors have developed a major work of medicinal/pharmaceutical chemistry using 3D-structural data elaborated from the X-ray crystallographic study of the ATPase domains of TOP2A and TOP2B. Although I am not competent to judge the accuracy of their work (my skills are in pharmacology and genomics), I found the description of the results very convincing, explained with much details but very clearly, and with a number of pictures showing the interactions between candidate drugs and the catalytic sites of both TOP2A and TOP2B obtained through co-crystallography with the partner. The conclusion is that these drugs can, likely in an allosteric way, interact with TOP2B to inhibit ATP cleavage, but not with TOP2A. This rational approach brings more specificity than the empiric approach that drove to the development of dexrazoxane as a cardioprotective agent.

RESPONSE: We thank the Reviewer for positive comments and appreciation of our work.

Pharmacological aspects of TOP2B catalytic inhibition

The differences between the modus operandi of TOP2 poisons and TOP2 catalytic inhibitors are clearly explained. The approaches used to explore the differential activities on TOP2A and TOP2B of the first compound studied (“5c”, fig. 1d, e), as well as the optimised compound (topobexin, fig. 3b and ext. dat. fig. 7a) are quite relevant: ATP cleavage and DNA decatenation. (Incidentally, ext. dat. fig. 7g is labelled fig. 7f on the graph).

RESPONSE: We thank the Reviewer for pointing this out, and we have corrected the panel label in the revised manuscript.

The use of neonatal rat ventricular cardiomyocytes (NVCM) to evaluate the cardiac toxicity of the compounds studied simply via the release of LDH is certainly a valuable approach in a first attempt and has provided interesting data. It cannot be considered as specific however, since any kind of primary cells in culture would have probably given the same results. The results obtained on NVCM cells on the protection of daunorubicin-induced γ H2AX formation and on caspase activity are convincing, but the comparison of dexrazoxane to topobexin (ext. dat. fig. 8a) is not convincing and the differences in LDH release are not impressive, even if significant.

RESPONSE: Unlike many other *in vitro* models, the NVCM model used in the present study has been carefully developed and validated to provide the best available cardioprotective predictive value (both positive and negative) and we have previously shown it has very good concordance with the results of chronic *in vivo* cardiotoxicity experiments (e.g. Jirkovská-Vávrová *et al* Toxicol Res 2015, Kollárová-Brázdová *et al*. JPET 2020, Jirkovsky *et al*. Circ HF 2021). However, we acknowledge that the NVCM *in vitro* model is only predictive of cardioprotection and it cannot replace complex *in vivo* conditions of chronic cardiotoxicity experiments. As for the comparison of DEX and topobexin in this model, we only demonstrate that topobexin was able to show statistically significant cytoprotective effects at a lower concentration (0.1 μ M) than DEX which suggest its higher potency in these *in vitro* settings.

In the revised manuscript, we have edited the discussion text to specify that the comparison of topobexin and dexrazoxane was performed only *in vitro*:

“Furthermore, we have demonstrated that topobexin (9) is a cardioprotective molecule that is effective at lower concentrations than dexrazoxane *in vitro*, and highly effective at preventing AIC during anthracycline chemotherapy *in vivo*.”

For demonstrating the cardioprotective effect of a drug, an animal model, rabbit or rat, is much better than a cellular model. However, the protection against anthracycline-induced acute DNA damage (fig. 5b) is not in my view a reliable endpoint; what is important is the preservation of the contractility of the cardiac muscle on the long term, in other words the left ventricular systolic function. This has been determined by the authors by echocardiography and by systolic pressure measurements after catheterization: these two approaches are very reliable and are considered as highly predictive of cardiac toxicity (and protection) in the human clinical situation. The treatment protocol (10 weeks) appears quite satisfactory. The biochemical parameters (troponin T, BNP, etc.) are also useful but certainly less decisive than the maintenance of heart contractility.

RESPONSE: We thank the Reviewer for this positive assessment. Indeed, *in vitro* cardioprotection experiments served only for basic screening of the study compounds, and the acute exposure data in panel

5b demonstrates prevention of acute DNA damage, but a robust translational *in vivo* model, with appropriate examination of cardiac function, was necessary to demonstrate topobexin protection against chronic cardiotoxicity. We also agree that other individual parameters presented in this study are not as clear and convincing, but together they comprise multiple lines of evidence and, most importantly, they independently corroborate outcomes of the cardiac function examination regarding cardioprotection.

Discussion

I have appreciated the mention that the utilization of dexrazoxane in the clinical setting has remained limited because of the persistent belief that it interfered with anthracycline efficiency. This is absolutely true and the number of children who have not benefited of dexrazoxane and have developed disabling heart failure is sadly very high. The availability of a new drug, with a mechanism of action different from that of the bisdioxopiperazine drugs, may indeed modify the perception of cardiac protection of children receiving anthracyclines.

I have no other remarks concerning this manuscript which should be published without significant modifications.

RESPONSE: We thank the Reviewer for their suggestions and comments. The aim of providing cardiac protection to pediatric cancer patients is a particularly strong motivation for our group, and we are hopeful that we will be able to achieve this in the not too distant future. We now mention pediatric cancer survivors in the introduction, and have added the following comment to the discussion to help emphasize this point:

Topobexin has a mechanism of action that allows for isoform-selectivity and is distinct from that of the bisdioxopiperazine drugs, which may indeed modify perception and encourage greater utilization of pharmacological cardioprotection against AIC, particularly in pediatric cancer patients that will need a lifetime of healthy heart function after cancer treatment.